# Compartmentalised RNA catalysis in membrane-free coacervate protocells

Björn Drobot[1], Juan M. Iglesias-Artola [1], Kristian Le Vay [2], Viktoria Mayr[2], Mrityunjoy Kar[1], Moritz Kreysing [1], Hannes Mutschler [2] & T-Y Dora Tang [1]

Phase separation of mixtures of oppositely charged polymers provides a simple and direct route to compartmentalisation via complex coacervation, which may have been important for driving primitive reactions as part of the RNA world hypothesis. However, to date, RNA catalysis has not been reconciled with coacervation. Here we demonstrate that RNA catalysis is viable within coacervate microdroplets and further show that these membrane-free droplets can selectively retain longer length RNAs while permitting transfer of lower molecular weight oligonucleotides.

[1] Max-Planck Institute for Molecular Cell Biology and Genetics, Pfotenhauerstraße 108, 01307 Dresden, Germany. [2] Max-Planck Institute for Biochemistry, Am Klopferspitz 18, 82152 Martinsried, Germany. These authors contributed equally: Juan M. Iglesias-Artola, Kristian Le Vay. Correspondence and requests for materials should be addressed to M.K. (email: kreysing@mpi-cbg.de) or to H.M. (email: mutschler@biochem.mpg.de) or to T-Y.D.T. (email: tang@mpi-cbg.de)

ompartmentalisation driven by spontaneous self-assembly processes is crucial for spatial localisation and concentration of reactants in modern biology and may have been important during the origin of life. One route known as complex coacervation describes a complexation process[1,2] between two oppositely charged polymers such as polypeptides and nucleotides[3–7]. The resulting coacervate microdroplets are membrane free, chemically enriched and in dynamic equilibrium with a polymer poor phase. In addition to being stable over a broad range of physicochemical conditions, coacervate droplets are able to spatially localise and up-concentrate different molecules[3,8] and support biochemical reactions[9,10].

It has been hypothesised that compartments which form via coacervation could have played a crucial role during the origin of life by concentrating molecules and thus initiating the first biochemical reactions on Earth[11]. Coacervation has also been implicated in modern biology where it has been shown that the formation of membrane-free compartments or condensates such as P-bodies or stress granules within cells are driven by this mechanism[12,13]. These membrane-free organelles are chemically isolated from their surrounding cytoplasm through a diffusive phase boundary, permitting the exchange of molecules with their surroundings[14]. In addition, these compartments may localise specific biological reactions and play important roles in cellular functions such as spatial and temporal RNA localisation within the cell[15–18].

Whilst there is increasing evidence for the functional importance of RNA compartmentalisation via coacervation in modern biology, this phenomenon would also have been vitally important during a more primitive biology. Up-concentration and localisation could have enabled RNA to function both as a catalyst (ribozyme) and storage medium for genetic information, as required by the RNA world hypothesis[19]. To date, ribozymes have been encapsulated within eutectic ice phases[20,21] and protocell models such as water–oil-droplets for directed evolution experiments[22–24], membrane-bound lipid vesicles[25–27], and membrane-free compartments based on polyethelene glycol (PEG)/dextran aqueous two-phase systems (ATPS)[28]. Interestingly, RNA catalysis within ATPS exhibits an increased rate of reaction as a result of the increased concentration within the dextran phase. Despite these examples, RNA catalysis has not been demonstrated within coacervate-based protocells. Therefore, coacervate protocells based on carboxymethyl dextran sodium salt (CM-Dex) and poly-L-lysine (PLys) (Supplementary Fig. 1) were chosen as the model system due to their proven ability to encapsulate and support complex biochemical reactions catalysed by highly evolved enzymes[10]. In contrast to these enzymes, structurally simple ribozymes, which are thought to have played a key role during early biology, are prone to fold into inactive conformations in the absence of RNA chaperones or additional auxiliary elements[29–33], and therefore may be rendered inactive by interactions within the highly charged and crowded interior of coacervate microdroplets. Herein, we directly probe the effect of the coacervate microenvironment on primitive RNA catalysis, and show the ability of the coacervate microenvironment to support RNA catalysis whilst selectively sequestering ribozymes and permitting transfer of lower molecular weight oligonucleotides.

## Results

**Hammerhead activity in bulk coacervate phase.** We developed a real-time fluorescence resonance energy transfer (FRET) assay (see Methods) to investigate the effect of the coacervate microenvironment on catalysis of a minimal version of the hammerhead ribozyme derived from satellite RNA of tobacco ringspot virus (HH-min)[34]. HH-min and its FRET substrate (Fig. 1a,

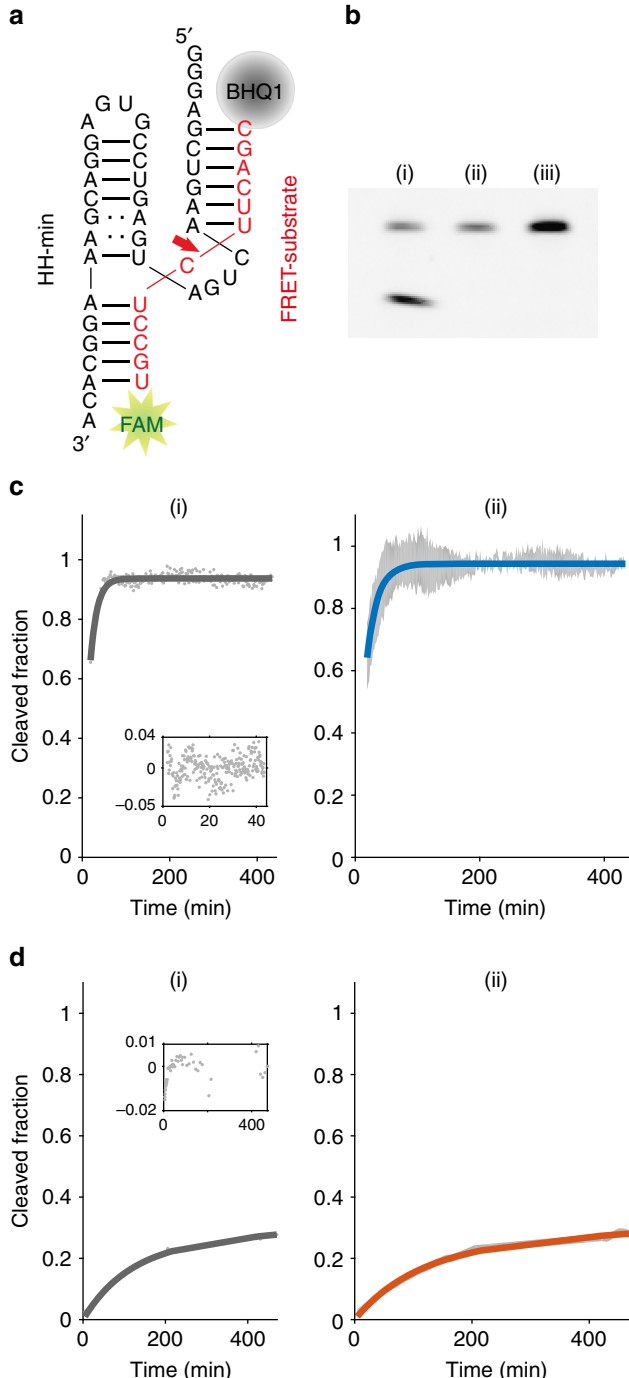

Methods) were incubated within a bulk polysaccharide/polypeptide coacervate phase or within coacervate microdroplets under single turnover conditions (see Methods). Cleavage of the FRET-substrate strand by HH-min increases the distance between 6-carboxyfluorescein (FAM) and Black Hole quencher 1 (BHQ1), resulting in increased fluorescence intensity. We further developed an inactive control ribozyme (HH-mut) by introducing two point mutations at the catalytic site (see Methods).

HH-min (1 μM) and FRET substrate (0.5 μM) were incubated within the CM-Dex : PLys bulk coacervate phase (4:1 final molar concentration, pH 8.0). The RNA was then separated from the coacervate phase and analysed by denaturing gel electrophoresis (see Methods, Fig. 1). Excitingly, fluorescence gel imaging showed the presence of cleavage product in the bulk coacervate phase

**Fig. 1** Cleavage of the FRET substrate under different conditions. **a** HH-min (black) and the FRET substrate (red). **b** Gel electrophoresis of RNA cleavage in bulk coacervate phase (CM-Dex : PLys, 4:1 final molar ratio); 0.5 μM of FRET substrate was incubated with 1 μM of (i) HH-min, (ii) HH-mut or (iii) no ribozyme in bulk phase (25 °C, 60 min). Samples were analysed by denaturing PAGE followed by fluorescence imaging. The lack of in-gel quenching of the FRET substrate likely results from modifications of BHQ1 during PAGE[51]. **c** Real-time cleavage kinetics in 10 mM Tris-HCl pH 8.3 and 4 mM MgCl$_2$. (i) A monoexponential fit (Methods, Eq. 3) (grey line) to kinetic data (grey dots) and residuals of the fit (inset); (ii) mean of the individual fits to each experiment (Blue line) with the standard deviation of the mean of the fits (grey data points) ($N = 5$). **d** Cleavage in bulk coacervate phase (normalised to the amount of cleaved product at $t = 530$ min from gel electrophoresis). (i) Biexponential fit (Methods, Eq. 4) (dark grey line) to experimental data (grey dots) with the residuals (inset); (ii) mean biexponential fit (orange) of individual fits ($N \geq 5$). Grey data points represent the standard deviation ($N = 5$) from the experimental data

containing HH-min (Fig. 1b). In contrast, control experiments in the absence of HH-min or in the presence of HH-mut showed no evidence of the cleavage product, confirming that the wild-type ribozyme drives substrate cleavage in the bulk coacervate phase (Fig. 1b). The FRET assay was further exploited to characterise the enzyme kinetics in both the bulk coacervate phase and buffer by time-resolved fluorescence spectroscopy under single turnover conditions by direct loading of HH-min and FRET substrate into either cleavage buffer or bulk coacervate phase (see Methods). The increase in fluorescence intensity of FAM was measured over time and normalised to the amount of cleaved product generated. Fitting the kinetic profiles of substrate cleavage in buffer conditions with a single exponential revealed an apparent rate constant, $k_0$ of $0.6 \pm 0.1$/min (Fig. 1c, $N = 5$), which was comparable to the $k_0$ obtained in buffer analysed by gel electrophoresis ($0.38 \pm 0.05$/min) (see Methods, Supplementary Fig. 2, $N = 6$) and to $k_{cat}$ values previously determined for a range of hammerhead ribozymes (0.01–1.5/min)[34,35]. In comparison, RNA cleavage within the bulk coacervate phase was clearly biphasic (Supplementary Fig. 3A) with an observed faster rate constant, $k_1$, of $1.0 \times 10^{-2} \pm 0.1 \times 10^{-2}$/min and a second slower rate constant, $k_2$, $1.5 \times 10^{-4} \pm 0.8 \times 10^{-4}$/min (errors obtained from 12 individual droplets). Thus, the fastest rate constant $k_1$ is 60-fold slower than in buffer conditions ($k_0 = 0.6 \pm 0.1$/min) indicative of reduced activity within the coacervate phase. In addition, the transition to biphasic kinetics within the coacervate phase compared to the aqueous buffer phase describes a different kinetic mechanism of HH-min within the coacervate phase (Fig. 1d). This may be attributable to heterogeneous ribozyme populations with variations to secondary structure and/or alternative conformational and equilibrium states, as observed for some HH systems in aqueous buffer conditions[36,37]. Circular dichroism (CD) spectra show a reduction in molar ellipticity ([θ]) for HH-mut within bulk coacervate phase compared to aqueous buffer with a small commensurate shift in the peak maxima from 265 nm to 268 nm, respectively (Supplementary Fig. 4). These results show that the fold of HH-mut is altered in the polyelectrolyte-rich bulk coacervate phase, with an overall loss of secondary structure that could affect catalytic activity. In addition, it is possible that the charged and crowded coacervate microenvironment restricts substrate binding, sterically hinders substrate–enzyme complex formation and/or spatially restricts diffusion of the cleavage assay components. Indeed, measured diffusion coefficients of HH-min tagged with TAMRA (TAM-HH-min) ($1.0 \pm 0.2$ μm$^2$/s) and FAM-substrate ($1.6 \pm 0.1$ μm$^2$/s) in the bulk coacervate phase (Fig. 2, mean and standard

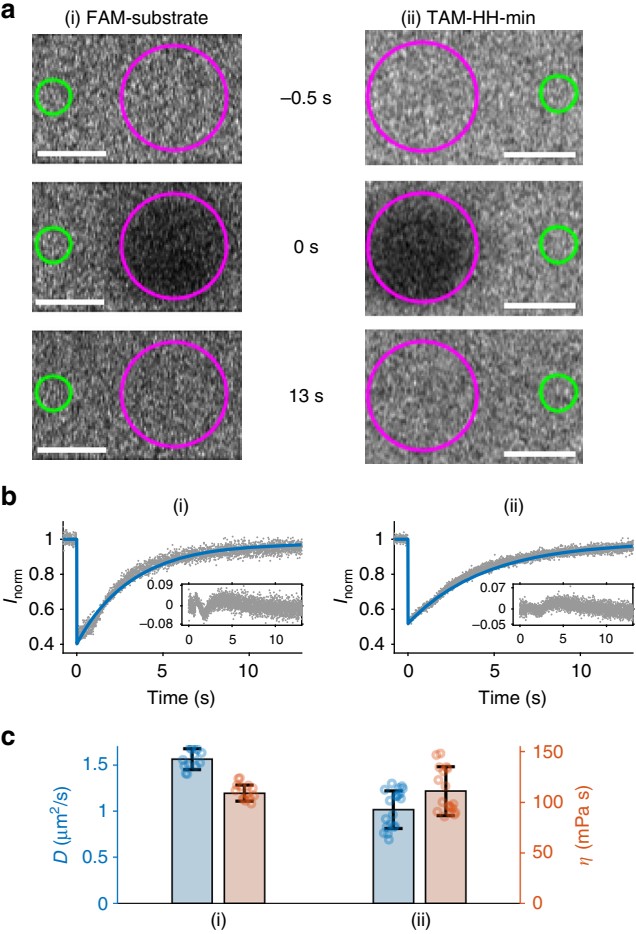

**Fig. 2** FRAP of bulk coacervate phase. Bulk coacervate phase (CM-Dex:PLys (4:1 final molar ratio) containing either (i) 0.36 μM FAM-substrate or (ii) 0.36 μM TAM-HH-min. **a** Output frames from confocal imaging (63×) are shown at $t = -0.5$ s before bleaching, directly after bleaching (magenta circle, $t = 0$ s) and $t = 13$ s after bleaching. The fluorescence intensity was normalised against a reference (green circle) and fit to standard equations. Scale bars are 5 μM. **b** Plots of normalised FRAP data for HH-min (ii) and FAM-substrate (ii) show the standard deviation (grey, $N = 10$) and fit (blue) from the same bleach spot radius. **c** Diffusion coefficients and viscosities obtained from **b**. Mean and standard deviations are from at least two different samples with analysis from ≥14 bleach spots for each experiment

deviations are from at least two different samples with analysis from at least 14 bleach spots from each experiment) from Fluoresence Recovery after Photobleaching (FRAP) analysis showed a significantly slower molecular diffusion of the ribozyme and substrate compared to predicted diffusion coefficients of RNA in buffer (~150 μm$^2$/s, Fig. 2)[38,39]. The decreased mobility is indicative of a more viscous and spatially restricted environment in the interior of the coacervate phase ($\eta = 114 \pm 21$ mPa. s, Fig. 2c).

**Hammerhead activity in coacervate microdroplets.** To test the activity of the ribozyme within individual droplets, the bulk coacervate phase containing ribozyme and substrate was re-dispersed in supernatant to produce microdroplets in solution (see Methods). The final concentration of enzyme and substrate in the microdroplet dispersion, formed from 1 μl of bulk coacervate phase redispersed in 49 μl of supernatant was equivalent to the final concentration of the bulk coacervate phase i.e. within

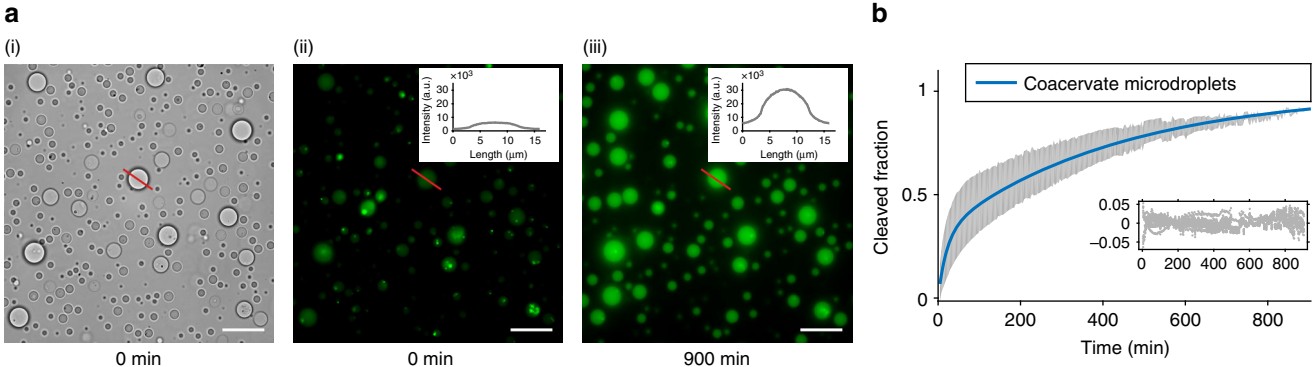

**Fig. 3** RNA catalysis in coacervate microdroplets. **a** (i) Wide-field optical microscopy images of CM-Dex:PLys (4:1 final molar ratio) coacervate microdroplets prepared in cleavage buffer (1 µM of HH-min and 0.5 µM FRET substrate). Fluorescence microscopy images at $t = 0$ min (ii) and $t = 900$ min (iii) show an increase in FAM fluorescence (see inset). Scale bars are 20 µM. **b** Background corrected and volume/endpoint normalised fluorescence intensity of droplets. Standard deviation of kinetics from 12 micro-droplets (grey) with the mean biexponential fit (blue) and residuals (inset)

50 µl of phase under single turnover conditions (1 µM of HH-min and 0.5 µM FRET substrate). Fluorescence optical microscopy images showed an increase in FAM fluorescence intensity in the droplets after 900 min (Fig. 3a).

Fitting the biphasic fluorescence signal allowed a direct comparison of kinetic profiles between the coacervate microdroplet (Fig. 3b, Supplementary Fig. 3B) and bulk coacervate phase environments. A modestly faster rate constant ($k_1$ and $k_2$) was observed in the microdroplets ($k_1$ of $4.4 \times 10^{-2} \pm 1.3 \times 10^{-2}$/min, $k_2$ of $2.3 \times 10^{-3} \pm 0.2 \times 10^{-3}$/min, $N = 12$) compared to the bulk coacervate phase ($k_1$ of $1.0 \times 10^{-2} \pm 0.1 \times 10^{-2}$/min, $k_2$ of $1.5 \times 10^{-4} \pm 0.8 \times 10^{-4}$/min, $N = 11$) (Supplementary Table 3, Supplementary Fig. 5). Determination of the partition coefficients of both the ribozyme and substrate ($K_{\text{HH-min}} = 9600 \pm 5600$ ($N = 12$) and $K_{\text{HH-substrate}} = 3000 \pm 2000$ ($N = 20$), Supplementary Fig. 6) by fluorescence spectroscopy (see Methods) showed that both TAM-HH-min and FAM-substrate partition strongly into the coacervate environment. 1 µl of bulk coacervate phase was used to prepare coacervate microdroplet suspensions in a total volume of 50 µl compared to 50 µl of bulk coacervate phase for bulk experiments. Thus, based on the measured partition coefficients, we calculated concentrations of 49.6 µM HH-min and 24.3 µM substrate in a single microdroplet, compared to 1 µM HH-min and 0.5 µM substrate within the bulk coacervate phase. Whilst the observed rate differences between the bulk coacervate and coacervate microdroplet phases could be due to variations in viscosity, quantitative FRAP analysis with two different FAM-substrate concentrations (0.36 µM and 24.3 µM) showed that the measured viscosities of the bulk coacervate phase and coacervate microdroplet are comparable within error (Supplementary Fig. 7). Therefore, secondary effects arising from the increased RNA concentration within the coacervate microdroplet phase may be responsible for the increased rate constants observed. The ribozyme and substrate concentrations are approximately 50 times more in the coacervate microdroplet (49.6 µM and 24.3 µM, respectively) compared to the bulk coacervate phase (1 µM and 0.5 µM, respectively). The difference in diffusion length scales of RNA in the microdroplet environment could lead to increased saturation of the ribozyme and therefore greater apparent rate constants in the coacervate microdroplets compared to bulk coacervate phase. In addition, enrichment of RNA could lead to changes in material properties such as water activity or dielectric constant, which have a direct impact on the rate of hammerhead-catalysed RNA cleavage[40,41]. Thus, secondary effects that result from the increased RNA concentration within coacervate microdroplets may concomitantly contribute to an increase in the apparent rate constant.

**Selective RNA partitioning into coacervates**. To further investigate the six-fold difference in the partition coefficient between the ribozyme and substrate, we characterised the differences in the rate and extent of sequestration of TAM-HH-min and FAM-substrate from the surrounding aqueous phase into the droplet after whole-droplet photobleaching. Coacervates containing FAM-substrate (12-mer) showed complete fluorescence recovery within 100 s and a recovery half time ($\tau$) of $22 \pm 3.5$ s ($N = 20$). In comparison, TAM-HH-min showed only 70% recovery after 300 s with $\tau = 189 \pm 14$ s ($N = 11$), attributed to either a low concentration of HH-min within the surrounding aqueous phase, a slow rate of transfer into the coacervate droplet from its exterior and/or immobilised ribozyme in the coacervate droplet, which is unable to exchange with RNA in the surrounding aqueous phase (Supplementary Fig. 8). The results from the FRAP experiments complement the equilibrium partition coefficient. The 12-mer substrate, with a lower partition coefficient ($K = 3000 \pm 2000$, $N = 20$) shows a faster exchange between the droplet and the surrounding aqueous phase compared with the 39-mer ribozyme, which has a higher partition coefficient ($K = 9600 \pm 5600$ ($N = 12$)), shows slower exchange of RNA with the surrounding environment. A strong correlation between FRAP half-lives and partitioning coefficients was also described for RNAs in other coacervate systems[4].

To investigate additional sequence-dependent effects on partitioning, we compared the partition coefficients of different 12-mer RNAs (Supplementary Fig. 6): FAM-substrate is pyrimidine rich but unstructured RNA; FAM-tet is a pyrimidine-rich hairpin structure with a stable UUCG tetraloop; FAM-flex is an unstructured purine-rich sequence (see Supplementary Table 2). Our results show that for unstructured RNAs an increase in purine content reduces partitioning of 12-mer RNAs (FAM-substrate vs. FAM-flex) approximately 10-fold. Likewise, we observe a decrease in the partition coefficient with an increase in secondary structure for pyrimidine-rich RNA (FAM-substrate vs. FAM-tet) (Supplementary Fig. 9). Thus, our results, and others[4,42,43], show that RNA sequestration and localisation within coacervate droplets is dependent on the length, sequence and also structure of the sequestered RNA.

For the RNAs specific to our HH ribozyme assay, the general selective retention of longer length polynucleotides with transfer of shorter length RNA can have interesting implications for ribozyme catalysis within coacervate droplets. To investigate this, we directly observed the localisation of TAM-HH-min and FRET-substrate. CM-Dex:PLys coacervate micro-droplets (4:1 final molar ratio) containing TAM-HH-min were loaded into one end of a capillary channel (Fig. 4a, region 1) while droplets

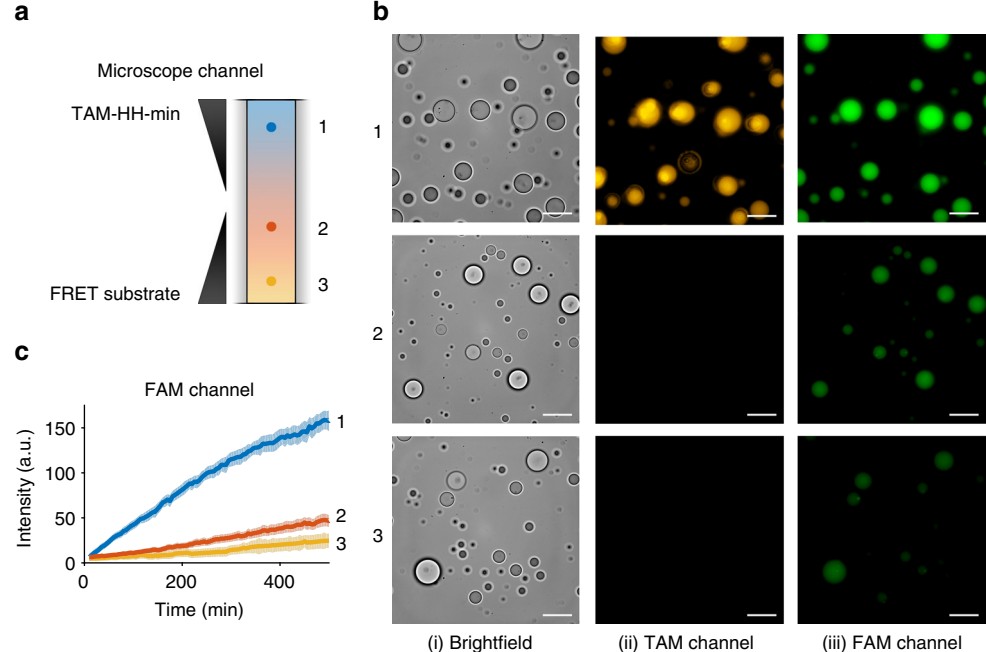

**Fig. 4** Localisation and retention of RNA within coacervate droplets. **a** Schematic of localisation experiments where CM-Dex:PLys (4:1 final molar ratio) coacervate droplets containing 0.36 μM (final concentration) TAM-HH-min were loaded into one end of a capillary channel (1). Droplets containing 0.36 μM FRET-substrate were loaded into the other end of the channel (3). **b** Wide-field optical microscopy images obtained using a 100 × oil immersion lens in (i) bright field and fluorescence mode using filters for (ii) TAM or (iii) FAM. Images were captured in regions 1, 2 and 3 at t = 500 min (scale bar: 20 μM). **c** FAM fluorescence intensity. Shaded regions represent the standard deviation of at least seven droplets

containing the FRET-substrate were loaded into the other end of the channel (Fig. 4a, region 3) in such a way as to prevent droplet mixing whilst permitting passive diffusion of molecules through the length of the channel (see Methods). Time-resolved fluorescence optical microscopy images in both the TAM and FAM channels were obtained at various locations along the capillary channel (Fig. 4a, regions 1, 2, 3). Imaging over 500 min in the TAM channel showed that, within the measurable resolution, no diffusion of the ribozyme to droplets in other regions of the channel occurs (Fig. 4b, region 2 and Supplementary Fig. 10). Conversely, over the time course, droplets in all three regions exhibited increased FAM fluorescence with droplets in region 1 with the highest intensity and droplets in regions 2 and 3 with comparatively lower intensity (Fig. 4c). Analysis of time-resolved images showed a delayed increase in the onset of cleaved product fluorescence in region 2 and a further delayed onset in cleaved product in region 3. These results are commensurate with diffusion of the FRET-substrate out of the droplets in region 3 and into droplets containing TAM-HH-min in region 1 where cleavage takes place. The cleaved product then diffuses out of the active droplets and into droplets in regions 2 and 3. Control experiments probing the transfer of FAM-substrate only show increased fluorescence intensity in regions 2 and 3 giving a direct confirmation that the 12-mer substrate diffuses between droplets (Supplementary Fig. 10). Taken together, our results show that longer length RNA (HH-min) is retained and spatially localised within the highly charged and crowded interior of the coacervate droplet, while shorter RNAs transfer between the droplets.

As other studies have shown that RNA rapidly exchanges from PLys and adenosine triphosphate (ATP, Supplementary Fig. 1) coacervate microdroplets into the surrounding environment[44], we also tested this coacervate system for selective localisation of RNA. To this end, localisation experiments were undertaken as previously described (see Methods) with PLys:ATP coacervate

microdroplets (4:1 final molar ratio) at pH 8: droplets containing either TAM-HH min or FAM-substrate were loaded into one end of a capillary channel, and coacervate droplets containing HH-mut were loaded into the other end of the capillary channel in such a way as to prevent droplet mixing (Supplementary Fig. 11). Fluorescence optical microscopy images obtained in the middle of the channel (Supplementary Fig. 11B, region 2) showed no change in the fluorescence intensity of TAM-HH-min (Supplementary Fig. 11C, i) over the course of the experiment (500 min). In contrast, a small increase in the fluorescence intensity from FAM-substrate (Supplementary Fig. 11C, ii) was observed after 300 min, suggesting a higher exchange rate of the 12-mer with the environment. Both the partition coefficient and whole droplet FRAP results for PLys:ATP coacervate microdroplets, containing either TAM-HH-min (39-mer), FAM-substrate (12-mer) or cleaved FAM-substrate (6-mer) (Supplementary Figs. 12, 13), confirmed a consistent trend in RNA retention based on RNA length with an order of magnitude difference in τ between the different oligonucleotides (Supplementary Fig. 12). Whilst the general trends are consistent with those observed with CM-Dex: PLys coacervate microdroplets, a direct comparison of whole droplet FRAP recovery times (Supplementary Table 4) between the two microenvironments shows that RNA has a stronger tendency to localise within the PLys:ATP microdroplets compared to CM-Dex:PLys microdroplets. The longer recovery times after whole droplet photobleaching in the PLys:ATP droplets could be attributed to differences in molecular interactions of RNA within the two microenvironments. For example, there may be increased RNA–PLys interaction in the PLys:ATP environment compared to the PLys:CM-Dex, where presence of CM-Dex could shield PLys–RNA interactions. In addition, favourable Pi–Pi stacking interactions between the aromatic rings of ATP and RNA would favour interactions of RNA within the PLys:ATP system. Taken together, the results show that membrane-free droplets prepared via coacervation offer general features such as

length-dependent RNA localisation and transfer. Moreover, our results also show that the strength of oligonucleotide selectivity is dependent on the composition of the coacervate microdroplets and the molecular sequence of RNA.

## Discussion

Here, we demonstrate that coacervate microdroplets offer intriguing properties for compartmentalised RNA catalysis. Our results show that these membrane-free microenvironments support RNA catalysis and up-concentrate oligonucleotides within their interiors. This effect is coupled to selective retention and release of RNA without additional energy input. These features could have been significant on early Earth where concentrations of RNA and their building blocks may have been low. Moreover, maintenance of the genetic identity of coacervate protocells could be achieved via spatial localisation of RNA catalysis and RNA genomes with spread of RNA building blocks or short genetic polymers between droplets. Whilst this work represents a key step in reconciling primitive RNA catalysis with selective protocellular compartmentalisation, it should also be noted that these features of compartmentalisation are significant in modern biology. To this end, there are immediate questions to be addressed. For example, how does the microdroplet environment alter the ribozyme mechanistic pathway and effect nucleotide selectivity into the droplet? How can we alter the physical chemistry of the droplets to further affect oligonucleotide selectivity? In addition, our experiments have focused on a nucleolytic ribozyme; however, rather than RNA cleavage, an increase in genetic and molecular complexity of RNA, e.g., by ligation would have been important during early Earth. Therefore, probing RNA synthesis through ligase activity within coacervate microdroplets will further contribute to the understanding of the role of coacervation on early Earth and modern biology.

## Methods

**Materials**. Trizma base (Tris), CM-Dex sodium salt (10–20 kDa, monomer: 162.14 g/mol), PLys hydrobromide (4–15 kDa, monomer: 161.67 g/mol), ATP disodium salt (551.14 g/mol), sodium hexametaphosphate ((NaPO$_3$)$_6$, 611.77 g/mol), formamide (CH$_3$NO, 45.04 g/mol), ethylenediaminetetracetic acid disodium salt dihydrate (EDTA, C$_{10}$H$_{14}$N$_2$Na$_2$O$_8$·2H$_2$O, 372.24 g/mol), fluorescein isothiocyanate isomer I (FITC, C$_{21}$H$_{11}$NO$_5$S, 389.38 g/mol), bromophenol blue (C$_{19}$H$_{10}$Br$_4$O$_5$S, 669.96 g mol$^{-1}$) and Orange G (C$_{16}$H$_{10}$N$_2$Na$_2$O$_7$S$_2$, 452.37 g/mol) were purchased from Sigma Aldrich and used without further purification. Magnesium chloride hexahydrate (MgCl$_2$·6 H$_2$O, 203.30 g/mol), sodium hydroxide (NaOH, 39.997 g/mol), sodium chloride (NaCl, 58.44 g/mol), urea (CH$_4$N$_2$O, 60.06 g/mol), hydrochloric acid (HCl, 37%, 36.46 g/mol) and boric acid (H$_3$BO$_3$, 61.83 g/mol) were purchased from Merck and used without further purification. Ammoniumperoxodisulfate (APS, (NH$_4$)$_2$S$_2$O$_8$, 228.20 g/mol), Rotiphorese® Gel 40 (Acrylamide 37, 5:1 Bisacrylamide) and Tetramethylethylendiamine (TEMED, C$_6$H$_{16}$N$_2$, 116.21 g/mol) were purchased from Roth. 5(6)-Carboxyte-tramethylrhodamine succinimidyl ester (TAMRA, C$_{29}$H$_{25}$N$_3$O$_7$, 527.53 g/mol) and SYBR gold stain (10,000× concentrate in DMSO) were purchased from Thermo Fisher Scientific. Oligo length standards (10/60 single strand RNA) was purchased from Integrated DNA technologies.

All DNA and tagged RNA oligonucleotides were synthesised by Eurofins Ebersberg, Germany, and used without further purification (Supplementary Table 2). Stocks of the RNA constructs were prepared in nuclease-free water and stored at −80 °C until use.

Nuclease-free water was purchased from Ambion and used to prepare aqueous stock solutions of CM-Dex (1 M, pH 8), PLys (0.2 M, pH 8), ATP (0.1 M, pH 8) and stored in the freezer (−20 °C) until use. The pH of the stock solutions was adjusted using NaOH, and concentrations were determined from the molecular weight of the monomer. Aqueous stock solutions of Tris-HCl (1 M, pH 8/pH 8.3) and MgCl$_2$ (1 M) were prepared and used for buffer solutions. Capillary channel slides for microscopy were custom made using PEGylated cover slips (22 × 22 mm) and microscope slides (25 mm × 75 mm).

**RNA synthesis**. A minimal, *trans*-acting hammerhead ribozyme (HH-min) derived from satellite RNA of tobacco ringspot virus and complementary substrate were produced by modification of the helix 1 hybridising arm in a *cis*-acting system[45]. An inactive variant (HH-mut) was produced by the introduction of two point mutations, G5A and G12A, which inhibit correct ribozyme folding and active

site protonation−deprotonation events, respectively[46]. The wild-type and inactive hammerhead ribozymes were transcribed in vitro by T7 RNA polymerase. The DNA templates for transcription were produced by fill-in of DNA oligonucleotides using GoTaq (Promega). Complementary pairs of DNA oligonucleotides contained the ribozyme gene (sTRSV_min_wt_TX/ sTRSV_min_mut_TX) and T7 promoter (5T7) (Supplementary Table 1). The double-stranded templates were purified using a Monarch PCR DNA Cleanup Kit (NEB, Biolabs, USA). Ribozyme RNA was transcribed from the double-stranded templates using the MEGAshortscript™ T7 Transcription Kit (ThermoFisher), and purified using RNeasy (Qiagen).

**Preparation of coacervates containing RNA HH and substrate**. Preparation of bulk coacervate phase and coacervate microdroplets containing RNA HH and substrate. Aqueous dispersions of CM-Dex:PLys coacervate microdroplets (4:1 molar ratio) or ATP:PLys (4:1 molar ratio) in 10 mM Tris-HCl pH 8.0 and 4 mM MgCl$_2$ were first prepared by mixing 200 μl of 1 M CM-Dex, 250 μl of 0.2 M PLys, 50 μl of 1 M Tris-HCl pH 8.0 and 20 μl of 1 M MgCl$_2$ and made up to 5 ml with nuclease-free water. Aqueous dispersions of PLys:ATP coacervate dispersions (4:1 molar ratio) in 10 mM Tris-HCl pH 8.0 and 4 mM MgCl$_2$ were produced by mixing 1000 μl of 0.2 M PLys, 500 μl of 0.1 M ATP, 50 μl of 1 M Tris-HCl pH 8.0 and 20 μl of 1 M MgCl$_2$ and made up to 5 ml with nuclease-free water. To produce a polymer-only bulk coacervate phase (approximately 100 μl), the aqueous dispersion of microdroplets was centrifuged (10 min, 4000×*g*) and the supernatant removed. To produce bulk coacervate phase containing RNA, RNA was added directly to 80 μl of bulk coacervate phase. The samples were mechanically mixed and centrifuged (10 min, 4000×*g*) and any excess water from the RNA stock solutions was removed. To produce coacervate microdroplet dispersions containing the same total concentration of RNA compared to bulk coacervate phase, 1 μl of bulk coacervate phase loaded with RNA (e.g., 50 pmol HH-min) was prepared as previously described. This 1 μl of the bulk coacervate loaded with RNA was made up to 50 μl with supernatant and vortexed to produce a dispersion of RNA loaded coacervate microdroplets. Final concentrations of RNA are calculated from the total volume, i.e., bulk coacervate samples contain 50 μl of bulk coacervate phase whilst the volume of the microdroplet dispersions accounts for both the volume of coacervate bulk phase and the supernatant it is dispersed in. Therefore, whilst the total concentrations are equivalent, there is approximately 50× less coacervate phase within the dispersion (1 μl) compared to the bulk coacervate phase (50 μl).

**Hammerhead ribozyme FRET assay**. A minimal trans-cleaving version of the tobacco ring spot virus hammerhead ribozyme (HH-min) was chosen as the model system for this study. To characterise the activity of HH-min, we employed a FRET assay. Cleavage of the FRET-substrate strand by HH-min increases the distance between FAM and BHQ1, resulting in increased fluorescence intensity. In addition, a modified version of the HH-min was developed as an inactive control ribozyme (HH-mut) by introducing two point mutations at the catalytic site. This mutant permits binding of the substrate, but is catalytically inactive.

**Hammerhead activity with gel electrophoresis**. Substrate cleavage assays were carried out in cleavage buffer (10 mM Tris-HCl, pH 8.3, and 4 mM MgCl$_2$) at 25 °C under single turnover conditions, i.e., with stoichiometric amounts of ribozyme and substrate; in this case, the ribozyme concentration is 2× the concentration of substrate. An excess of HH-min (1 μM) compared to FRET-substrate (0.5 μM) were mixed in RNAse-free water, the reaction was initiated by the addition of cleavage buffer. Aliquots were taken at varying time points and quenched on ice by the addition of four volumes of RNA gel loading buffer (formamide, EDTA (10 mM, pH 8.0), bromophenol blue (0.025% w/v). The substrate and cleavage products were separated by denaturing PAGE on 20% acrylamide gels and run in 1× TBE buffer. Bands were visualised by FAM-tag fluorescence (Typhoon FLA-5000, GE Healthcare Life Sciences, $\lambda_{ex}$ = 473 nm, $\lambda_{em}$ = 520 nm), and the intensities of cleavage product at each time point were determined by integration of band intensities using ImageQuant.

The first-order time constant $\tau$ was obtained by fitting the cleaved and uncleaved kinetic profiles to single exponential fits (Eqs. 1, 2, respectively), which were globally optimised.

$$I(t)^{cleaved} = I_{max}^{cleaved} - I_{max}^{cleaved} \times e^{-\left(\frac{t}{\tau}\right)} + C, \qquad (1)$$

$$I(t)^{uncleaved} = C + \left(I_{max}^{uncleaved} - C\right) \times e^{-\left(\frac{t}{\tau}\right)}. \qquad (2)$$

$I(t)$ is the band intensity, $C$ is an offset and $\tau$ the first-order time constant. The first-order rate constant ($k_0$) is then determined by $1/\tau$.

To obtain the differences in fluorescence quantum yield for the cleaved and uncleaved substrate (FRET effect), the maximum intensities at the start and the end points ($I_{max}$) were obtained and their ratio determined.

**Gel electrophoresis of hammerhead activity in coacervates**. HH-min and FRET-substrate at final concentrations of 1 μM and 0.5 μM, respectively, were

loaded into CM-Dex:PLys (4:1 final molar ratio) bulk coacervate phase for single turnover conditions, i.e., with stoichiometric amounts of ribozyme and substrate; in this case, the ribozyme concentration is $2\times$ the concentration of substrate. RNA was extracted from 5 μl of CM-Dex:PLys (4:1 final molar ratio) bulk coacervate phase (25 °C) or from 5 μl of CM-Dex:PLys (4:1 final molar ratio) coacervate microdroplet dispersions after 900 min (25 °C) by sequential addition of 5 μl 1 M NaCl (final concentration 4.8 mM), 5 μl of 1.25 M hexametaphosphate (final concentration −6.0 mM) and 90 μl RNA loading buffer (final volume—83%) containing EDTA (final concentration—8 mM) and Orange G and 10 s of vortexing and 1 s of centrifugation between each addition. The reaction mixture was heated at 80 °C for 10 min, centrifuged and placed on ice for at least 5 min. 10 μl of the reaction mixture was loaded into a pre-run 20% polyacrylamide gel and then run at 300 V in 1× TBE buffer until the dye had run to the bottom of the gel. The gel was imaged using Typhoon FLA-9500, GE Healthcare Life Sciences, with $\lambda_{ex} = 473$ nm and $\lambda_{em} = 520$ nm. Band intensities were measured using ImageQuant at a specific time point and uncleaved substrate was corrected by the FRET effect factor (1.69) (Supplementary Fig. 12). The fraction of cleaved substrate of the total sum of cleaved and uncleaved substrate was determined and used to normalise kinetic data obtained from spectroscopy or microscopy at specific time points. Gel electrophoresis was undertaken with FRET-substrate and FAM-substrate with a single-strand RNA molecular marker on a pre-run 20% polyacrylamide gel and run at 15 W in 1× TBE buffer. The gel was stained with SYBR gold and imaged using a Typhoon FLA-5000, GE Healthcare Life Sciences, with $\lambda_{ex} = 473$ nm and $\lambda_{em} = 520$ nm. Uncropped gels are shown in Supplementary Figure 15.

**Kinetics of ribozyme activity within bulk environments.** The enzyme reaction was incorporated into either buffer or bulk coacervate phase by adding the appropriate volume from ribozyme stock (HH-min or HH-mut) solutions into the bulk coacervate phase, CM-Dex:PLys (4:1 final molar ratio) to achieve a final concentration of 1 μM. The reaction was initiated by adding FRET-substrate at a final concentration of 0.5 μM. After mixing the reaction mixture, 20 μl of sample was loaded into a 384-well plate (microplate, PS, Small Volume, LoBase, Med. binding, Black, Greiner Bio-one). The enzyme activity was monitored using TECAN Spark 20 M well plate reader spectrophotometer (Tecan AG, Männedorf, Switzerland) by measuring the increase in FAM fluorescence over time ($\lambda_{exc} = 485$ nm and $\lambda_{em} = 535$ nm, 10 nm bandwidth, at 25 °C). The fluorescence intensity of HH-mut was used as the background intensity and subtracted from HH-min data. This was then normalised by determining the amount of FRET-substrate cleaved by gel electrophoresis at the endpoint of the experiments as described previously. Kinetic profiles were fit to either single exponential growth (Eq. 3) under buffer conditions or bi-exponential growth (Eq. 4) for bulk coacervate experiments.

$$I(t) = 1 - e^{-\left(\frac{t-t_0}{\tau}\right)}, \tag{3}$$

$$I(t) = 1 - \left(A_1 e^{-\left(\frac{t-t_0}{1}\right)} + (1-A_1)e^{-\left(\frac{t-t_0}{2}\right)}\right), \tag{4}$$

where $I(t)$ is the normalised intensity, $A_1$ is the amplitude of the first and $1-A_1$ the amplitude of the second exponential, $t$ is the time in min, $t_0$ is the dead time between sample preparation and the first measurement (separately measured), $\tau_1$ or $\tau_2$ are the fitted time constants. The corresponding rate constants $k_1$ or $k_2$ are obtained from $\tau_1$ or $\tau_2$ where $k=1/\tau$.

**Hammerhead catalysis within coacervate microdroplets.** FRET-based HH-min activity within CM-Dex:PLys coacervate droplets (4:1 final molar ratio) was undertaken by addition of the FRET-substrate into a dispersion of coacervate droplets containing ribozyme, prepared from 1 μl of bulk coacervate phase loaded with 50 pmol HH-min in 49 μl of supernatant. Final concentration under single turnover conditions were 1 μM for HH-min and 0.5 μM FRET-substrate. Dispersions were loaded into custom-made PEGylated channel capillary slide for microscope imaging. Time-resolved bright field and fluorescent images of the droplets were obtained using a 100× oil immersion objective (Plan-Apochromat 100×/1.40 Oil DIC, Zeiss) mounted onto a Zeiss Axiovert 200M inverted widefield microscope equipped with a 16-channel CooLED pE-300-W and an ANDOR ZYLA fast sCMOS camera. Image acquisition was controlled with the Metamorph software (1 frame/min, 100 ms exposure time) for 900 min. Images were taken with Chroma filter set comprised of $\lambda_{exc} = 470 \pm 20$ nm (Batch number: 111753) and $\lambda_{em} = 525 \pm 25$ nm (Batch number: 112298) and dichromatic mirror with $\lambda = 495$ LP with pixel dimension of 0.0631 μm² and image bit depth of 16 and analysed using the Fiji software to obtain the integrated fluorescence intensities divided by the volume of the droplet as a function of time for 12 coacervate microdroplets. The amount of substrate cleaved was normalised by gel electrophoresis at a given time point as described previously. Kinetic parameters were derived from fitting the kinetic data to Eq. 4. To test for reproducibility, the experiment was repeated with a different batch of HH-min using the same experimental conditions (Supplementary Fig. 5).

**Fluorescence recovery after photobleaching.** Fluorescence recovery after photobleaching was undertaken within both CM-Dex:PLys and PLys:ATP coacervate microdroplets (4:1 final molar ratio) and the bulk coacervate phase (CM-Dex:PLys 4:1 final molar ratio) containing either TAM-HH-min (0.36 μM), FAM-substrate (0.36 μM), FAM-substrate (24.3 μM) or FAM-cleaved substrate (0.36 μM). Samples were prepared as previously described and loaded into capillary slides mounted in a Zeiss LSM 880 inverted single point scanning confocal microscope equipped with a 32 GaAsP PMT channel spectral detector and a 32-channel Airy Scan detector and imaged using a 63× oil immersion objective (Plan-Apochromat 63× 1.4 Oil DIC, Zeiss). Bleaching was achieved by additional excitation with a 405 nm laser diode and 355 nm DPSS laser, an Argon Multiline Laser produced the excitation wavelength of $\lambda_{FAM} = 488$ nm or $\lambda_{TAM} = 561$ nm and emission wavelengths $\lambda_{FAM} = 479–665$ nm (Laser line blocking pin at 488 nm) or $\lambda_{TAM} = 562–722$ nm. Imaging time varied depending on the region of interest but was typically between 12 ms/frame and 100 ms/frame.

The fluorescence intensity as a function of time for the bleached area, reference and the background was obtained using FIJI and the recovery of the bleached region was normalised against the background and the reference region for either bulk CM-Dex:PLys (4:1 final molar ratio) coacervate or PLys:ATP (4:1 final molar ratio) coacervate experiments. An additional normalisation for droplet-based FRAP was undertaken by dividing the fluorescence by the fluorescence of the whole droplet[47,48]. The kinetic profiles were fit to Eq. 5 using MATLAB to obtain the time constant, $\tau$, of fluorescence recovery or of transport into the droplet (whole droplet FRAP).

$$I(t) = \begin{cases} 1 & \text{for } t < t_0 = 0\,\text{s} \\ 1 - A \times \exp\left(-\frac{t-t_{\text{bleach}}}{\tau}\right) & \text{for } t \geq t_0 = 0\,\text{s} \end{cases} \tag{5}$$

where $t_0 = 0$ s is defined as the first time point after bleaching. The diffusion coefficient is related to the time constant $\tau$ by the relation[49,50]. The diffusion coefficient was averaged over 20 bleaching events across at least two different samples:

$$D = 0.88 \frac{r^2}{4\log(2)}. \tag{6}$$

Here $r$ is the radius of the bleached spot. The diffusion coefficient was averaged over 20 bleaching events across at least two different samples. Interpretation of the apparent diffusivity from photobleaching experiments may be complicated due to the complexity of the liquid coacervate phase including interactions between the studied fluorescence labelled RNA and the polymers. However, by approximating the coacervate phase as a Newtonian fluid, an estimation of the viscosity via the Stokes–Einstein relation (Eq. 7) could be made.

$$\eta = \frac{k_B T}{6\pi R_h D}, \tag{7}$$

where $\eta$ the viscosity (mPa s), $k_B$ the Boltzmann constant (m² kg/s²/K), $T$ the temperature in K and $R_h$ the hydrodynamic radius (in m) calculated from length of the RNA using Eq. 8 (ref. [39]):

$$R_h = (5 +/- 0.28)10^{-10}N^{(0.38 +/- 0.01)}, \tag{8}$$

where $N$ is the length in nucleotides.

**Ribozyme activity and localisation in coacervate droplets.** To determine the localisation of the RNA, CM-Dex:PLys (4:1 final molar ratio) or PLys:ATP (4:1 final molar ratio) coacervate micro-droplets emulsions (1 μl bulk coacervate phase mixed with 49 μl supernatant) containing TAM-HH-min (0.36 μM) were loaded into one end of a capillary channel whilst droplets containing the FRET-substrate (0.36 μM) were loaded into the other end of the capillary channel. Control experiments also included loading droplets containing either TAM-HH-min or FAM-substrate into one end of the channel whilst microdroplets containing HH-mut were loaded into the other end of the capillary channel.

Time-resolved bright field and fluorescent images of the droplets along different parts of the imaging channel were obtained using a 100× oil immersion objective (Plan-Apochromat 100×/1.40 Oil DIC, Zeiss) mounted onto a Zeiss Axiovert 200M inverted widefield microscope equipped with a 16-channel CooLED pE-300-W and an ANDOR ZYLA fast sCMOS camera. Images in the TAM channel were taken with $\lambda_{exc} = 542 \pm 13.5$ nm (AHFanalysentechnik AG, Batch number: 116338-116340) and $\lambda_{em} = 593 \pm 23$ nm (AHF analysentechnik AG, Batch number: 116448–116450) with laser beam splitter H560 LPXR superflat (AHF analysentechnik AG, Batch number: 6-4209) and in the FAM channel with Chroma filter set comprised of $\lambda_{exc} = 470 \pm 20$ nm (Batch number: 111753) and $\lambda_{em} = 525 \pm 25$ nm (Batch number: 112298) and dichromatic mirror with $\lambda = 495$ LP. Image acquisition was controlled with the Metamorph software (5 min/frame, 100 ms exposure time) for 500 min or 90 min for the controls with pixel dimension of 0.0631 μm² and image bit depth of 16 and analysed using the Fiji software. The integrated fluorescence intensities divided by the volume of the droplet as a function of time were obtained for at least 12 coacervates microdroplets.

**Fluorescence-based partitioning of RNA**. TAM-HH-min, FAM-substrate, FAM-cleaved product, FAM-Tet, FAM-Flex, FITC and TAMRA were loaded into 150 µl dispersions containing 3 µl of bulk coacervate phase (CM-Dex:PLys 4:1 molar ratio) to achieve an initial concentration of 0.5 µM ($c_{ini}$). After an equilibration time of 10 min, the coacervate phase (3 µl) was separated from the supernatant (147 µl) by centrifugation (10 min at 10,000×g). To ensure both fractions were treated equally, the 147 µl supernatant fraction was filled to a final volume of 150 µl ($V_{ini}$) by the addition of 3 µl of RNA-free bulk CM-Dex:PLys (4:1 final molar ratio) coacervate phase (supernatant phase), whilst the 3 µl of RNA-containing bulk 4:1 CM-Dex:PLys (4:1 final molar ratio) coacervate phase fraction was made up to 150 µl ($V_{ini}$) with RNA-free supernatant (coacervate phase). Coacervates of both samples were dissolved by the addition of 10 µl NaCl (5 M). The Fraction, $F$, of RNA within either the bulk CM-Dex:PLys (4:1 final molar ratio) coacervate phase or the supernatant phase was obtained using Eq. 9:

$$F_{phase} = \frac{I_{phase}}{\sum I}, \tag{9}$$

where the $I_{phase}$ is the fluorescence intensity of either the coacervate or supernatant phase measured from 20 µl of sample using the TECAN spark 20M of either FAM-substrate ($\lambda_{exc}^{FAM} = 485$ nm, $\lambda_{em}^{FAM} = 535$ nm) or TAM-HH-min ($\lambda_{exc}^{TAM} = 544$ nm, $\lambda_{em}^{TAM} = 576$ nm) and $\sum I$ is the sum of both.

The corresponding concentrations in both phases were calculated using Eq. 10:

$$c_{phase} = \frac{F_{phase} \times c_{ini} \times V_{ini}}{V_{phase}}, \tag{10}$$

where $c_{ini}$ is the initial RNA concentration (0.36 µM), $V_{ini}$ is the initial volume (150 µl) and $V_{phase}$ is the volume of either the supernatant phase (147 µl) or the bulk coacervate phase (3 µl).

Based on these concentrations, the partition coefficient $K$ was calculated using Eq. 11:

$$K = \frac{c_{coac}}{c_{super}}, \tag{11}$$

where the $c_{coac}$ is the concentration of RNA in the bulk coacervate phase and $c_{super}$ is the concentration of RNA in the supernatant phase.

**Ribozyme activity in supernatant phase**. In order to estimate the contribution of FRET-substrate cleavage in the aqueous surrounding of the coacervate microdroplets, cleavage experiments were performed in the supernatant using a HH-min concentration of 0.007 µM (based on the partitioning coefficient for the ribozyme) and 0.5 µM FRET-substrate concentration (maximum possible concentration). The data were compared to a control in supernatant under single turnover conditions (1 µM HH-min, 0.5 µM FRET-substrate). The kinetics of the substrate cleavage was obtained using TECAN Spark 20M well plate reader spectrophotometer (Tecan AG, Männedorf, Switzerland) by measuring the increase in FAM fluorescence over time ($\lambda_{exc} = 485$ nm and $\lambda_{em} = 535$ nm, 10 nm bandwidth, at 25 °C).

**Circular dichroism**. To investigate the secondary structure of the hammerhead ribozyme, HH-mut was mixed into either buffer ($c_{HH} = 1$ or 2 µM) or bulk coacervate phase ($c_{HH} = 2$ µM) (CM-Dex:PLys, 4:1 molar ratio) and loaded into a 1-mm Special Quartz Cuvette (200 µl). CD spectra were measured using a Chirascan™-Plus CD Spectrometer (Applied Photophysics), with data collected every 1 s at 25 °C from 320 to 200 nm with a resolution of 1 nm. Either 5 or 10 repeat spectra were measured for buffer and bulk coacervate samples, respectively. Background spectra of buffer alone and bulk coacervate phase were taken under the same conditions and were subtracted from the spectra of HH-mut in buffer and bulk coacervate phase, respectively. All spectra were offset at 320 nm, normalised for the cuvette pathlength and converted from $\Delta A$ to molar ellipticity ([$\theta$]) (deg cm²/dmol).

## Data availability:

Data supporting the findings of this manuscript are available from the corresponding authors upon reasonable request. Computer code can be downloaded at https://doi.org/10.17617/1.6J.

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

## Acknowledgements
Funding was provided by the Volkswagen Foundation and the MaxSynBio consortium, which is jointly funded by the Federal Ministry of Education and Research of Germany and the Max Planck Society. We thank the light microscopy facility (LMF) for their continuous support and guidance during the FRAP experiments, and Jeff Oegema and Ian Henry at the Scientific computing facility (MPI-CBG) for support with making the data available. We acknowledge the Boehringer Ingelheim Fonds for the award of a PhD fellowship to J.M.I.-A. We thank Andre Nadler and Carl Modes for useful discussion.

## Author contributions
H.M., M.K., and T.-Y.D.T. conceived the project; T.-Y.D.T., H.M., B.D., J.M.I.-A., K.L.V., M.Kar and V.M. undertook the experiments and data analysis; H.M., T.-Y.D.T., B.D., J.M.I.-A., K.L.V. and M.Kar discussed the results; T.-Y.D.T., H.M., B.D. and K.L.V. wrote the manuscript.

## Additional information

**Competing interests:** The authors declare no competing interests.

