## [Peer Review File · Nature Communications]

Reviewers' comments:

Reviewer #1 (Remarks to the Author):

Review – Compartmentalized RNA catalysis in membrane – free coacervate protocells

Manuscript ID: NCOMMS-18-06329

When composing your report, the following questions might assist you in writing an incisive, well-justified review. What are the major claims of the paper? Are they novel and will they be of interest to others in the community and the wider field? If the conclusions are not original, it would be helpful if you could provide relevant references. Is the work convincing, and if not, what further evidence would be required to strengthen the conclusions? On a more subjective note, do you feel that the paper will influence thinking in the field? Please feel free to raise any further questions and concerns about the paper.

We would also be grateful if you could comment on the appropriateness and validity of any statistical analysis, as well the ability of a researcher to reproduce the work, given the level of detail provided.

General Comments:

This paper describes the first report of ribozyme activity within a complex coacervate environment. This result is of particular interest in the context of the RNA world hypothesis for the origin of life. The authors show preferential partitioning and retention of longer RNA (ribozymes) into coacervate droplets as compared with shorter substrate strands. These results provide support for the idea that liquid-liquid phase separated droplets could serve as a method for compartmentalizing material and providing a competitive evolutionary advantage.

Specific Comments:

1. Figure S2A shows the kinetic analysis for the reaction performed in a bulk coacervate phase. However, the data plateaus at a cleaved fraction value of approximately 0.4, while the data in Figure S3B continues up to a value of 0.7. However, the data for microdroplet experiments extend to a cleaved fraction of ~1.0. Why is such a difference observed?

My first thought was that the kinetic measurements might be inaccurate in this case due to diffusional limitations. However, taking the values of the diffusivity determined via FRAP (1-2 $\mu\text{m}^2/\text{s}$), both the ribozyme and substrate should be able to diffuse on the order of 300-400 μm over the course of a 1000 min experiment. This should be sufficient for equilibration across the coacervate phase. Could the limitation instead be diffusion throughout the entire sample since the droplet case would be much more dispersed throughout the sample volume, while the bulk phase would be more spatially localized in the tube? Even if the samples were mixed during the experiment, there should be a significant difference in the diffusional profiles of these samples. If true, this diffusive limitation could account for the differences in the kinetic rate parameters obtained between the bulk and droplet phases, rather than differences in partitioning of the TAM-HH-min and FAM-substrate into a large vs. a small volume of the same material.

2. While I agree that the positive result of RNA catalysis inside of a coacervate phase is exciting, the wording in the manuscript almost makes it seem like such results were also unexpected. Is there some reason why this might be true? Previous studies on protein-based enzymes have shown the ability to retain and possibly enhance enzyme activity in coacervate phases. Is there some aspect of RNA-based materials that might cause this trend not to hold for these materials?
3. I assume that the use of a capillary diffusion experiment to test for material exchange between the droplets was performed in order to avoid coalescence events that would complicate the analysis. It might be worth mentioning this in the manuscript. However, I do wonder if it would have been beneficial to use the type of microfluidic technology that has been reported previously

(van Swaay *et al.*, *Angewandte Chemie*, (2015), **54**, 8398-8401) to prepare stable dispersions of droplets and thus highlight the potential for observing a direct competitive advantage.

4. I suggest the following two references to further support the authors case about length-dependent competitive sequestration of polymers into a coacervate:
 - Zhang and Shklovskii, *Physica A*, (2005), **352**, 216-238.
 - Peng and Muthukumar, *Journal of Chemical Physics*, (2015), **143**(24), 243133.
5. A tremendous amount of information about the materials and procedures is available in the SI. It would be very useful to include a statement in the materials and methods section (and throughout the manuscript where relevant) describing the types of information available in the SI.
6. Much of the work in this manuscript was done in “single-turnover conditions.” Does this simply mean that a stoichiometric amount of ribozyme and reactant were present in the mixture, or that each ribozyme was only capable of performing a single reaction and did not recover into a “ground state?” In the SI the concentrations appear to be 2:1 ribozyme:substrate. Please clarify.

Minor Comments:

1. Please define abbreviations like CM-Dex and PLys in the manuscript.
2. At the bottom of Page 3 of the SI, “than” should be “then”.
3. At the top of Page 4 of the SI there appears to be a glitch in the following sentence: “TAM-HH-min or FAM-substrate were loaded into 150 μ l dispersions containing into 3 μ l of bulk coacervate phase achieve a final initial concentration of 0.36 μ M (C_{ini}).”
4. What is the error in the data shown in Figure S1B?
5. In the caption for Figure S2B “plot” should be “plotted”.

Reviewer #2 (Remarks to the Author):

This paper addresses an interesting question, whether ribozyme activity is altered in coacervates, and whether ribozymes can maintain genetic identity in the coacervate phase. The authors cite the potential interest in coacervates as model protocells (as in the title), which motivates these experiments.

The main results are 1) the HH-min ribozyme works in the coacervate phase but is quite a bit slower than in aqueous buffer, 2) the HH-min ribozyme kinetics are biphasic in the coacervate phase, 3) these rates are modestly increased in droplets compared to bulk coacervate, 4) the HH-min ribozyme does not diffuse much among droplets but the shorter substrate RNA does.

Overall, my impression is that while the questions and results are interesting, more work and discussion is needed to understand the results, especially in the context of prior work. There are two main issues. First, the work (result 4) must be taken in context of Jia, Hentrich, and Szostak, 2014, which studied diffusion of RNA among coacervate droplets and concluded that exchange was rapid. The studies presented here must reconcile their opposing results with the previous study. For example, if the difference is the composition of the coacervate, the authors could validate their experimental setup by repeating the composition of Jia *et al.* in their microscopic droplet mixing experiment. Along this vein, the first instance of “CM-Dex:PLys” should be defined. I was assuming this represents a carboxylated dextran and lysine; if so, the reason for choosing this composition should be given since it is not an obvious prebiotic choice. The composition should be justified in any case.

Second, the observations of different rate constants and biphasic behavior are presented phenomenologically in the manuscript (results 1-3). I believe more attention, at least discussion, and experiments if possible, regarding possible mechanisms for each of these observations is needed to understand the potential importance of these results to the field. For example, a statistically significant acceleration of ~ 5 -fold in rate constants was observed in microdroplets vs. bulk phase. The text states a 'slight alternation of the material properties' but the mechanism of such an effect, even if only proposed but not verified, is not obvious to me. Do the authors mean the viscosity? If so, this could probably be measured. A similar concern relates to the biphasic curve (some mention is given of alternate conformers - perhaps this could be tested by chemical probing or even spectroscopically, e.g., CD). An explanation for the quite significant slowing of the ribozyme in the coacervate is also needed given the observation in cited literature of a hammerhead ribozyme accelerated in an ATPS.

Some additional comments are given below.

Pg 2: What exactly does kick starting mean in this context?

Page 3: I am not sure if this 60-fold (?) difference should be called 'an order of magnitude'; it would be better to simply give the fold-difference.

Figure 1C: The curve fit equations should be given explicitly with coefficients and constants. Does the curve begin from 0? If not, why not? What is the dead time of this experiment? Is (i) showing a different experiment from (ii); if so, how are these different? How do the grey points in (ii) indicate standard deviation - why would the stdev increase over time?

Figure 1D: Again, the curve fit equation should be given. These curves appear to begin at 0, but the relative amplitude of the two exponential fits should be given. Also, a supplementary figure should be included to show this graph as a semilog plot to demonstrate the two regimes. Same question as Fig. 1C with respect to stdev.

Figure 3A: Axes labels on insets are needed. I assume these are distributions of intensity; some comment on the procedure for intensity normalization between samples is needed.

Why would FRAP recovery only reach 70% for HH-min? This needs at least a proposed explanation.

Semantic issue: I do not think 'protocell' is normally used to describe coacervates; this could potentially be confusing to readers who understand the word 'protocell' to be a membrane-bound compartment. I would recommend eliminating this word from the title.

Reviewer #3 (Remarks to the Author):

The study entitled "Compartmentalized RNA catalysis in membrane - free coacervate protocells" reported the ribozyme activity in coacervate and size-dependent diffusion of RNAs among coacervate microdroplets. The authors analyzed the kinetics of a ribozyme and diffusion rate in detail. The manuscript was clearly written and the measurement and analysis were well performed. However, I have two major concerns as described below and they are critically important for the significance of this study to readers in a broad field of science. Therefore, I cannot recommend this manuscript for publication in Nature Communications.

Major points

1. The first concern is about novelty. I agree that ribozyme activity has not been observed in coacervate. But there have been several studies about other biochemical reactions, such as actinorhodin polyketide synthesis and gene expression, as the authors cited in references 8-9.

Therefore, ribozyme reaction in coacervate is not surprising and I don't believe this is a significant advance. I admit that the authors precisely analyzed the kinetics and diffusion and this study is worth publishing, but a more specific journal would be suitable.

2. The second concern is about one of the main claim of this study, selective retention of a long RNA in coacervate. I understood that this claim based on the result of Figure 4. I think there are two problems in this experiment to withdraw the conclusion.

First, the localization of the substrate RNA was not directly measured. I agreed that the HH-RNAs were localized in region 1, whereas the localization of the substrate RNA was not directly determined. Instead, the authors observed the FRET signal produced by ribozyme activity, which depends on the localization of both ribozyme and substrate and thus cannot be regarded as a direct evidence of substrate localization. Direct measurement of the substrate localization should be measured to conclude the different retention time.

Second, the difference in the retention of the two RNAs can be attributed to other factors than RNA sizes, such as labeled fluorescent compound (TAM or FAM), RNA sequences, or RNA secondary structures. To verify the effect of RNA size, additional experiments with various size and sequence of RNAs are required.

Minor point

The subsection of "Preparation of bulk coacervate phase..." of Materials and Method section, was difficult to understand. I believe the problems is lack of explanation of several phases: coacervate phase, polymer phase, bulk polymer phase, supernatant phase. Is the "coacervate phase" same as polymer phase? Clarification of these terms would be kind for readers.

Reviewer #1 (Remarks to the Author):

General Comments:

This paper describes the first report of ribozyme activity within a complex coacervate environment. This result is of particular interest in the context of the RNA world hypothesis for the origin of life. The authors show preferential partitioning and retention of longer RNA (ribozymes) into coacervate droplets as compared with shorter substrate strands. These results provide support for the idea that liquid-liquid phase separated droplets could serve as a method for compartmentalizing material and providing a competitive evolutionary advantage.

Specific Comments:

1. Figure S2A shows the kinetic analysis for the reaction performed in a bulk coacervate phase. However, the data plateaus at a cleaved fraction value of approximately 0.4, while the data in Figure S3B continues up to a value of 0.7. However, the data for microdroplet experiments extend to a cleaved fraction of ~1.0. Why is such a difference observed? My first thought was that the kinetic measurements might be inaccurate in this case due to diffusional limitations. However, taking the values of the diffusivity determined via FRAP (1-2 $\mu\text{m}^2/\text{s}$), both the ribozyme and substrate should be able to diffuse on the order of 300-400 μm over the course of a 1000 min experiment. This should be sufficient for equilibration across the coacervate phase. Could the limitation instead be diffusion throughout the entire sample since the droplet case would be much more dispersed throughout the sample volume, while the bulk phase would be more spatially localized in the tube? Even if the samples were mixed during the experiment, there should be a significant difference in the diffusional profiles of these samples. If true, this diffusive limitation could account for the differences in the kinetic rate parameters obtained between the bulk and droplet phases, rather than differences in partitioning of the TAM HH-min and FAM-substrate into a large vs. a small volume of the same material.

We thank the reviewer for the comment regarding the system being diffusion limited within the coacervate phase and microdroplet environment. To address this, we have calculated the concentration of ribozyme and substrate within the bulk coacervate phase and in the coacervate microdroplet (Shown in the table below) based on our partitioning experiments and observe an almost 50 x increase in the concentration of the ribozyme and substrate within the coacervate microdroplets compared to the bulk coacervate phase.

	[Ribozyme] (μM)	[Substrate] (μM)
Bulk coacervate phase	1	0.5
Coacervate microdroplets	49.6	24.3

Therefore, we attribute the difference in cleaved fraction at specific time points as observed in Figure S3A and 4B (previous Figure S2A and 3B) to a dilution effect or up-concentration effect within the bulk coacervate phase. It would be expected that with increased concentrations, as observed in other enzymatic reactions in solution, the distances of molecular diffusion will be reduced. This will lead to increased collision and binding events between the enzyme and substrate and increased rates of reaction. With the lower concentrations in the bulk coacervate phase, it would be expected that 100% cleavage will take place over a long time frame.

Therefore, the differences in the kinetic rate parameters are attributed to secondary effects from the up concentration of the enzyme and substrate and this has been described in the following text starting on page 5, line 122.

.... “ From the partition coefficient we calculated concentrations of 49.6 μM HH-min and 24.3 μM substrate in a single microdroplet compared to 1 μM HH-min and 0.5 μM substrate within the bulk coacervate phase. Whilst the observed rate differences between the bulk coacervate and coacervate microdroplet phases could be due to variations in viscosity, quantitative FRAP analysis with two different FAM-substrate concentrations (0.36 μM and 24.3 μM) showed that the measured viscosities of the bulk coacervate phase and coacervate microdroplet are comparable within error (Figure S7). Therefore, secondary effects arising from the increased RNA concentration within the coacervate microdroplet phase may be responsible for the increased rate constants observed. The ribozyme and substrate concentrations are approximately 50 times more in the coacervate microdroplet (49.6 μM and 24.3 μM respectively) compared to the bulk coacervate phase (1 μM and 0.5 μM respectively). The difference in diffusion length scales of RNA in the microdroplet environment could lead to increased saturation of the ribozyme and therefore greater apparent rate constants in the coacervate microdroplets compared to bulk coacervate phase. In addition, enrichment of RNA could lead to changes in material properties such as water activity or dielectric constant, which have a direct impact on the rate of hammerhead-catalysed RNA cleavage.^{35,36} Thus, secondary effects that result from the increased RNA concentration within coacervate microdroplets may concomitantly contribute to an increase in the apparent rate constant.”

New references have been included on page 24, lines 591-596

35. Nakano, S., Karimata, H. T., Kitagawa, Y. & Sugimoto, N. Facilitation of RNA Enzyme Activity in the Molecular Crowding Media of Cosolutes. *J. Am. Chem. Soc.* 131, 16881–16888 (2009).

36. Nakano, S.-I., Kitagawa, Y., Miyoshi, D. & Sugimoto, N. Hammerhead ribozyme activity and oligonucleotide duplex stability in mixed solutions of water and organic compounds. *FEBS Open Bio* 4, 643–650 (2014).

2. While I agree that the positive result of RNA catalysis inside of a coacervate phase is exciting, the wording in the manuscript almost makes it seem like such results were also unexpected. Is there some reason why this might be true? Previous studies on protein-based enzymes have shown the ability to retain and possibly enhance enzyme activity in coacervate phases. Is there some aspect of RNA-based materials that might cause this trend not to hold for these materials?

As the referee states correctly, enzyme catalysis and *in vitro* transcription and translation within the coacervate environment have been shown previously. However, in these examples, the enzymatic machineries are large and highly evolved macromolecules, which are adapted to fold and function in crowded environments. They have highly evolved molecular scaffolds and often auxiliary domains, which enable the stable formation of active sites and interfaces (depending on their function). In the case of primitive RNA catalysis (such as the minimal form of our hammerhead ribozyme), there are no or only very few stabilizing domains. Thus, the highly charged and crowded interior of the coacervate phase might very well disrupt or destabilize primitive ribozymes and thus inhibit their activity. Therefore, despite previous results, it would not be expected that minimal forms of ribozymes are active within the coacervate microenvironment.

Therefore, additional text has been added to the manuscript to explain why the result cannot be assumed.

On page 2, line 46, we have now written:

“Therefore, coacervate protocells based on Carboxymethyl Dextran sodium salt (CM-Dex) and Poly-L-Lysine (PLys) (Figure S1) were chosen as the model system due to their proven ability to encapsulate and support complex biochemical reactions catalysed by highly evolved enzymes^{9,10}. In contrast to these enzymes, structurally simple ribozymes, which are thought to have played a key role during early biology, lack structural stability and therefore may be rendered inactive by interactions within the highly charged and crowded interior. Herein, we directly probe the effect of the coacervate microenvironment on primitive RNA catalysis, and show the ability of the coacervate microenvironment to support RNA catalysis whilst”

References have been made to references 9 and 10 on page 21, line 536 and page 22, line 539

9. Crosby, J. et al. Stabilization and enhanced reactivity of actinorhodin polyketide synthase minimal complex in polymer–nucleotide coacervate droplets. *Chem. Commun.* **48**, 11832–11834 (2012).

10. Tang, T.-Y. D., van Swaay, D., deMello, A., Ross Anderson, J. L. & Mann, S. *In vitro* gene expression within membrane-free coacervate protocells. *Chem. Commun.* **51**, 11429–11432 (2015).

3. I assume that the use of a capillary diffusion experiment to test for material exchange between the droplets was performed in order to avoid coalescence events that would complicate the analysis. It might be worth mentioning this in the manuscript. However, I do wonder if it would have been beneficial to use the type of microfluidic technology that has been reported previously (van Swaay et al., *Angewandte Chemie*, (2015), **54**, 8398-8401) to prepare stable dispersions of droplets and thus highlight the potential for observing a direct competitive advantage.

The reviewer is correct and the capillary diffusion experiment was undertaken to test for material exchange. There are some advantages to using the microfluidic device as described in the 2015 Van Swaay paper i.e. droplets prepared by this microfluidic devices are stable to coalescence over a time frame of weeks and the droplets show increased monodispersity compared to droplets prepared by hand. However, there is no added value in using the microfluidic devices for the capillary diffusion experiments as the droplets prepared by hand are stable over the course of the experiment (600 mins) with no evidence of droplet coalescence after droplet loading

As the reviewer has suggested we have mentioned the use of the capillary experiments to allow for material exchange whilst avoiding coalescence. The sentence below was in the materials and methods but has now been moved to the main text:

“ in such a way as to prevent droplet mixing whilst permitting passive diffusion of molecules through the length of the channel.”

The main text now reads (page 6, line 162):

... “in such a way as to prevent droplet mixing whilst permitting passive diffusion of molecules through the length of the channel (see materials and methods)”

And the materials and methods (page 17, line 425) now reads:

.... “(1 μ l bulk coacervate phase mixed with 49 μ l supernatant) containing TAM-HH-min (0.36 μ M) were loaded into one end of a capillary channel whilst droplets containing the FRET-substrate (0.36 μ M) were loaded into the other end of the capillary channel. Control experiments ...”

4. I suggest the following two references to further support the authors case about length-dependent competitive sequestration of polymers into a coacervate:

- Zhang and Shklovskii, *Physica A*, (2005), 352, 216-238.
- Peng and Muthukumar, *Journal of Chemical Physics*, (2015), 143(24), 243133.

We thank the reviewers for these interesting papers and have included both references as new references 37 and 38 (see page 6, line 152 and page 24 lines 597-600).

37. Zhang, R. & Shklovskii, B. I. Phase diagram of solution of oppositely charged polyelectrolytes. *Phys. Stat. Mech. Its Appl.* **352**, 216–238 (2005).

38. Peng, B. & Muthukumar, M. Modeling competitive substitution in a polyelectrolyte complex. *J. Chem. Phys.* **143**, 243133 (2015).

5. A tremendous amount of information about the materials and procedures is available in the SI. It would be very useful to include a statement in the materials and methods section (and throughout the manuscript where relevant) describing the types of information available in the SI.

As recommended by the editor, all the materials and methods have been moved to the main manuscript. Additionally, on the reviewer’s advice, we have added in, where appropriate, additional references to the materials and method in the main text.

On page 3, lines 57-66:

“We developed a real-time fluorescence resonance energy transfer (FRET) assay (see materials and methods) to investigate the effect of the coacervate microenvironment on catalysis of a minimal version of the hammerhead ribozyme derived from satellite RNA of tobacco ringspot virus (HH-min)²⁹. HH-min and its FRET-substrate (Figure 1A, materials and methods) were incubated within a bulk polysaccharide / polypeptide coacervate phase or within coacervate microdroplets under single turnover conditions (see materials and methods). Cleavage of the FRET-substrate strand by HH-min increases the distance between 6-carboxyfluorescein (FAM) and Black Hole quencher 1 (BHQ1), resulting in increased fluorescence intensity. We further developed an inactive control ribozyme (HH-mut) by introducing two point mutations at the catalytic site (see materials and methods).”

On page 3, line 74:

“ The FRET assay was further exploited to characterize the enzyme kinetics in both the bulk coacervate phase and buffer by time resolved fluorescence spectroscopy under single turnover conditions by direct loading of HH-min and FRET-substrate into either cleavage buffer or bulk coacervate phase (see materials and methods)..”

On page 4, line 82:

which was comparable to the k_0 obtained in buffer analysed by gel electrophoresis ($0.38 \pm 0.05 \text{ min}^{-1}$) (see materials and methods, Figure S2)

On page 4, line 109:

“To test the activity of the ribozyme within individual droplets, the bulk coacervate phase containing ribozyme and substrate was re-dispersed in supernatant to produce microdroplets in solution (see materials and methods).”

On page 6, line 164:

“in such a way as to prevent droplet mixing whilst permitting passive diffusion of molecules through the length of the channel (see materials and methods)....”

On page 7, line 184:

“To this end, localization experiments were undertaken as previously described (see materials and methods) ...”

Amendments to main figures:

Figure 1- figure legend Page 26, line 633,

“(C) Real-time cleavage kinetics in 10 mM Tris•HCl pH 8.3 and 4 mM MgCl₂. (i) A monoexponential fit (materials and methods, Equation 3)”

6. Much of the work in this manuscript was done in “single-turnover conditions.” Does this simply mean that a stoichiometric amount of ribozyme and reactant were present in the mixture, or that each ribozyme was only capable of performing a single reaction and did not recover into a “ground state?” In the SI the concentrations appear to be 2:1 ribozyme:substrate. Please clarify.

Single turnover conditions means that there is stoichiometric amounts or an excess of enzyme. Under these conditions, the ribozyme would perform a single reaction and, in an ideal situation, would return to its ground state with full dissociation of the substrate/product.

To make this clear, we have included in the materials and methods on page 12, line 298-299 and on page 13 lines 322-323 the following text:

“i.e. with stoichiometric amounts of ribozyme and substrate, in this case the ribozyme concentration is 2x the concentration of substrate.”

Minor Comments:

1. Please define abbreviations like CM-Dex and PLys in the manuscript.

This has been defined in the main text and the materials and methods and an additional SI figure (Figure S1) – supplementary information (page, line 3-6) has been included to show the structures of the coacervate components.

Main text (page 2, last line 47):

“Carboxymethyl Dextran sodium salt (CM-Dex) and Poly-L-Lysine (PLys)”

Figure S1: Molecular structures of coacervate components Poly-L-Lysine (PLys), Carboxymethyl-dextran sodium salt (CM-Dex) and Adenosine 5' Triphosphate (ATP) and molecular structures of dye molecules, 5(6)-Carboxytetramethylrhodamine N-succinimidyl ester (TAMRA) and Fluorescein isothiocyanate isomer I (FITC/FAM).

Additionally, where appropriate, other abbreviations have been defined. The changes are shown below:

Page 4, line 100-101:

“Indeed, measured diffusion coefficients of TAM-HH-min ($1.0 \pm 0.2 \mu\text{m}^2\cdot\text{s}^{-1}$) and FAM-substrate ($1.6 \pm 0.1 \mu\text{m}^2\cdot\text{s}^{-1}$) in the bulk coacervate phase.”

To:

“Indeed, measured diffusion coefficients of HH-min tagged with TAMRA (TAM-HH-min)”

2. At the bottom of Page 3 of the SI, “than” should be “then”.

This has been changed.

3. At the top of Page 4 of the SI there appears to be a glitch in the following sentence: “TAM-HH-min or FAM-substrate were loaded into 150 μl dispersions containing into 3 μl of bulk coacervate phase achieve a final initial concentration of 0.36 μM (cini).”

We thank the reviewer for this. The sentence has been modified to account for the new and additional experiments. Pahe 17, line 445.

From:

“TAM-HH-min, FAM-substrate, FAM-cleaved product, FAM-Tet, FAM-Flex, FITC and TAMRA were loaded into 150 µl dispersions containing 3 µl of bulk coacervate phase (CM-Dex : PLys 4:1 molar ratio) to achieve a final initial concentration of 0.5 µM (c_{ini}).”

To:

“TAM-HH-min or FAM-substrate were loaded into 150 µl dispersions containing 3 µl of bulk coacervate phase to achieve a final initial concentration of 0.36 µM (c_{ini}).”

4. What is the error in the data shown in Figure S1B?

The error in the data for Figure S2B (previous Figure S1B) was obtained from 6 repeats of the experiment. Global fitting of the cleaved and uncleaved kinetic profiles to a single exponential fit yields 0.38 +/- 0.05 s⁻¹. The data for each time point and their errors are now shown in Figure S2B with the single exponential fit. In addition, the text has been modified in the main text to state the error and the materials and methods modified to describe the fitting procedure:

Main Text, Page 4, line 83:

comparable to the k₀ obtained in buffer analysed by gel electrophoresis (0.38 ± 0.05 min⁻¹) (see materials and methods, Figure S2).

Changes to materials and methods, Page 12, line 308:

“ The first order time constant τ was obtained by fitting the cleaved and uncleaved kinetic profiles to single exponential fits (Equations 1 and 2 respectively) and was optimised globally.

$$I(t)^{cleaved} = I_{max}^{cleaved} - I_{max}^{cleaved} \cdot e^{-\left(\frac{t}{\tau}\right)} + C \quad (\text{Equation 1})$$

$$I(t)^{uncleaved} = C + (I_{max}^{uncleaved} - C) \cdot e^{-\left(\frac{t}{\tau}\right)} \quad (\text{Equation 2})$$

I(t) is the band intensity, C is an offset and τ the first order time constant. The first order rate constant (k₀) is then determined by 1/τ.”

Amended Figure S1, now Figure S2 (Supplementary information page 4)

Figure S2: FRET assay for HH ribozyme in cleavage buffer (10 mM Tris, 4 mM MgCl₂) at 25 °C. **(A)** Characterisation was undertaken by gel electrophoresis and fluorescence gel imaging without HH-min and 0.5 μM FRET-substrate (i), 1 μM HH-min and 0.5 μM FRET-substrate (ii) or 1 μM HH-mut and 0.5 μM FRET-substrate at 5 min, 10 min, 20 min, 30 min and 60 min. The gels show HH-min is active whilst HH-mut is inactive. **(B)** The kinetics of HH-min were determined by integrating the band intensities of the cleaved and uncleaved bands and performing a global fit to a single exponential (Equation 1), yielding a first order rate constant of $0.38 \pm 0.05 \text{ min}^{-1}$ (see materials and methods). Data points are an average of six independent measurements. Note that the in-gel fluorescent intensities of cleaved and uncleaved FAM-labelled RNA are different and are attributed to the FRET effect (Figure S13B).

4. In the caption for Figure S2B “plot” should be “plotted”.

Figure S2 is now Figure S3 (found in supplementary information page 5). The grammar in the figure legend has been changed (supplementary information, page 5, line 82).

“The increase in fluorescence intensity from cleaved product (grey data) was plotted as a function of time and the thickness of the grey data points are from the standard deviation from five repeats.”

Reviewer #2 (Remarks to the Author):

This paper addresses an interesting question, whether ribozyme activity is altered in coacervates, and whether ribozymes can maintain genetic identity in the coacervate phase. The authors cite the potential interest in coacervates as model protocells (as in the title), which motivates these experiments.

The main results are 1) the HH-min ribozyme works in the coacervate phase but is quite a bit slower than in aqueous buffer, 2) the HH-min ribozyme kinetics are biphasic in the coacervate phase, 3) these rates are modestly increased in droplets compared to bulk coacervate, 4) the HH-min ribozyme does not diffuse much among droplets but the shorter substrate RNA does.

Overall, my impression is that while the questions and results are interesting, more work and discussion is needed to understand the results, especially in the context of prior work. There are two main issues. First, the work (result 4) must be taken in context of Jia, Hentrich, and Szostak, 2014, which studied diffusion of RNA among coacervate droplets and concluded that exchange was rapid. The studies presented here must reconcile their opposing results with the previous study. For example, if the difference is the composition of the coacervate, the authors could validate their experimental setup by repeating the composition of Jia et al. in their microscopic droplet mixing experiment.

We thank the reviewer for their comments and have added new experimental data regarding RNA localization with coacervates using the molecular composition from Jia et al. i.e. PLys: ATP. Our experiments, which probe different length RNA within PLys : ATP extends the study by Szostak et al., which shows that a RNA 15-mer will rapidly exchange between phases. We found the same general trends in selective RNA retention in PLys : ATP droplets compared to CM-Dex : PLys and direct comparisons between the two systems show that the same length RNA will transfer more rapidly into and out of CM-Dex : PLys compared to PLys : ATP. Consequently, we found that our hammerhead ribozyme (39-mer) also strongly co-localizes within PLys : ATP droplets (with only very slow exchange rates) compared to the more mobile 12-mer substrate. Thus, the PLys : ATP system also points towards the general possibility of stable co-localization of longer RNAs within coacervate protocells. We attribute the differences between the CM-Dex : PLys and PLys : ATP systems to variations in the molecular interactions of RNA with the coacervate components. Additional partitioning experiments show that partitioning of RNA is also dependent on the sequence of the RNA (new data and figure S9, supplementary information, page 11 and also see below). A new paragraph describing the additional findings have been included into the main text, new SI figures which summarises and compares the time constant of fluorescence intensity for CM-Dex : PLys and PLys : ATP from FRAP measurements. Additionally, the materials and methods have been changed to provide additional information for the new experiments.

An explanation into the main text has been included (page 7, line 181):

“As other studies have shown that RNA rapidly exchanges from PLys and Adenosine Triphosphate (ATP, Figure S1) coacervate microdroplets into the surrounding environment³⁹, we also tested this coacervate system for selective localization of RNA. To this end, localization experiments were undertaken as previously described (see materials and methods) with PLys : ATP coacervate microdroplets (4:1 molar ratio) at pH 8: Droplets containing either TAM-HH min or FAM-substrate were loaded into one end of a capillary channel, and coacervate droplets containing HH-mut were loaded into the other end of the capillary channel in such a way as to prevent droplet mixing (Figure S11). Fluorescence optical microscopy images

obtained in the middle of the channel (region 2, Figure S11B) showed no change in the fluorescence intensity of TAM-HH-min (Figure S11C, i) over the course of the experiment (500 min). In contrast, a small increase in the fluorescence intensity from FAM-substrate (Figure S11C, ii) was observed after 300 min, suggesting a higher exchange rate of the 12-mer with the environment. Whole droplet FRAP experiments of PLys / ATP coacervate microdroplets containing either TAM-HH-min (39-mer), FAM-substrate (12-mer) or cleaved FAM-substrate (6-mer) (Figure S12) confirmed a consistent trend in RNA retention based on RNA length with an order of magnitude difference in τ between the different oligonucleotides (Figure S12). Whilst the general trends are consistent with those observed with CM-Dex : PLys coacervate microdroplets, a direct comparison of whole droplet FRAP recovery times (Table S4) between the two microenvironments show that RNA has a stronger tendency to localize within the PLys : ATP microdroplets compared to CM-Dex : PLys microdroplets. The longer recovery times after whole droplet photobleaching in the PLys : ATP droplets could be attributed to differences in molecular interactions of RNA within the two microenvironments. For example, there may be increased RNA-PLys interaction in the PLys : ATP environment compared to the PLys : CM-Dex, where presence of CM-Dex could shield PLys - RNA interactions. In addition, favourable Pi-Pi stacking interactions between the aromatic rings of ATP and RNA would favour interactions of RNA within the PLys : ATP system. Taken together, the results show that membrane-free droplets prepared via coacervation offer general features such as length dependent RNA localization and transfer. Moreover, our results also show that the strength of oligonucleotide selectivity is dependent on the composition of the coacervate microdroplets and the molecular sequence of RNA.”

The new figures for the supplementary information are shown below:
 Supplementary information Page 3:

Table S4: Summary of the time constant (τ) of the fluorescence recovery of different length RNA from whole droplet FRAP of coacervate microdroplets prepared from a 4:1 molar ratio of either CM-Dex : PLys or PLys : ATP.

	TAM-HH-mut (39-mer)	FAM-HH-subs (12-mer)	FAM-Cleaved substrate (6-mer)
CM-Dex / PLys	189 +/-14 s	22 +/- 3.5	4.2 +/- 1.4 s
PLys / ATP	4800 +/- 4300 s	800 +/- 400 s	19 +/- 7 s

Supplementary information Page 13:

Figure S11: Localisation experiments for PLYs : ATP (4:1 molar ratio) coacervate microdroplets. (A) PLYs : ATP Coacervate microdroplets containing either 0.4 μM of TAM-HH-min (i) or FAM-substrate (ii) were loaded into one end of a capillary channel and PLYs : ATP coacervate microdroplets containing 0.4 μM HH-mut were loaded into the other end of the channel. Optical microscopy images were taken every 5 min for 500 min with $\lambda_{\text{exc}} = 542 \pm 13.5$ nm and $\lambda_{\text{em}} = 593 \pm 23$ nm (TAM-channel) or with $\lambda_{\text{exc}} = 469 \pm 20$ nm and $\lambda_{\text{em}} = 525.5 \pm 23.5$ nm (FAM-channel) for TAM-HH-min and FAM-substrate respectively. (B) Optical microscopy images from position 2 show no increase in fluorescence intensity after 500 min whilst optical microscopy images show a slight increase in fluorescence intensity of FAM-substrate in position 2 over time. (C) Plots of the mean integrated fluorescence intensities (solid line) and the standard deviation (shaded area) of at least 7 droplets as a function of time show no diffusion of TAM-HH-min through the length of the channel (i) whilst the increase in fluorescence intensity in region 2 shows diffusion of the FAM-substrate from region 1 to region 2 (ii) after 500 min.

Supplementary information Page 14:

Figure S12: Whole droplet FRAP of PLYs: ATP (4:1 final molar ratio) microdroplet dispersions containing either 0.5 μM (total concentration) TAM-HH-min (39-mer) (blue) (i) or 0.5 μM (total concentration) FAM-substrate (12-mer) (red) (ii) or 0.5 μM (total concentration) FAM-cleaved substrate (6-mer) (yellow) (iii) using confocal microscopy with $\lambda_{\text{FAM}} = 488$ nm or $\lambda_{\text{TAM}} = 514$ nm and emission wavelengths $\lambda_{\text{FAM}} = 479$ to 665 nm (laser line blocking pin at 488 nm) or $\lambda_{\text{TAM}} = 535$ to 704 nm. Bleaching was undertaken with additional 355 and 405 nm lasers. (A) Box plot of the recovery half-time (τ) shows a general trend of increasing τ with longer length RNA

indicative of less exchange for longer length RNA compared to shorter length RNA in PLys : ATP microdroplets; (i) 39-mer : $\tau = 4800 \pm 4300$ s ; (ii) 12-mer : $\tau = 800 \pm 400$ s; (iii) 6-mer : $\tau = 19 \pm 7$ s **(B)** Fluorescent Confocal microscopy images of whole droplet FRAP for 0.4 μ M FAM-cleaved substrate showing output frames at $t = -14$ s (before bleaching), $t = 0$ s (at bleaching) and $t = 100$ s. The output images show full recovery of FAM-cleaved substrate. Scale bar: 5 μ m. The fluorescence intensity as a function of time was obtained from the images from the bleached droplet (red), reference (green) and background (yellow). **(C)** Plot showing fluorescence recovery of the droplet after removal of the background and normalised to reference for TAM-HH-min (i) FAM-substrate (ii) and FAM-cleaved substrate (iii) the data was fit (solid line) to obtain τ . The large errors in τ for longer length RNA is attributed to fitting to a fluorescence recovery curves with incomplete recovery. The shaded region shows the data distribution from at least 6 droplets. Analysis of the data shows full recovery for the shortest length (6 mer) RNA whilst the longer length RNA (12 mer and 39 mer) showed less than 40 % recovery after 290 s.

Supplementary information Page 11:

Figure S9: Partitioning (K) of three different sequences RNA (i) 12-mer FAM-substrate (ii) 12-mer FAM-Flex (iii) 12-mer FAM-Tet. To obtain the partition coefficient 0.36 μ M (total concentration) of FAM tagged RNA was incubated in 150 μ l CM-Dex / PLys coacervate microdroplet solution (3 μ l bulk coacervate phase + 147 μ l supernatant) and left to equilibrate for 10 min. The dispersion was centrifuged and the supernatant phase separated from the coacervate phase. Either 3 μ L of the bulk coacervate phase or 147 μ l supernatant was made up to 150 μ L by the addition of 147 μ l RNA free supernatant or 3 μ l RNA free coacervate, respectively. The fractions of FAM tagged RNA in the coacervate phase and in the supernatant phase were obtained spectroscopically and the partitioning coefficients were calculated using Equation 7-9 and plotted with the y-axis on the logarithmic scale. Corresponding partitioning coefficients are (i) 12-mer FAM-substrate ($K = 3000 \pm 2000$, $n = 20$), (ii) 12-mer FAM-Flex ($K = 114 \pm 33$, $n = 9$), (iii) 12-mer FAM-Tet ($K = 126 \pm 10$, $n = 9$). The results show that the partitioning of RNA is not only dependent on its length but also on its sequence/structure.

Due to the additional experiments, the materials and methods have been modified, please see below.

In the materials on Page 10, line 234.

“Adenosine 5’ Triphosphate disodium salt (ATP, 551.14 g/mol)”

Page 11, line 273

“Aqueous dispersions of CM-Dex:PLys coacervate microdroplets (4:1 molar ratio) or ATP : PLys (4:1 molar ratio) in 10 mM Tris•HCl pH 8.0 and 4 mM MgCl₂ were first prepared by mixing 200 µl of 1 M CM-Dex, 250 µl of 0.2 M PLys, 50 µl of 1 M Tris•HCl pH 8.0, 20 µl of 1 M MgCl₂ and made up to 5 ml with nuclease free water. Aqueous dispersions of PLys:ATP coacervate dispersions (4:1 molar ratio) in 10 mM Tris•HCl pH 8.0 and 4 mM MgCl₂ were produced by mixing 1000 µl of 0.2 M PLys, 500 µl of 0.1 M ATP, 50 µl of 1 M Tris•HCl pH 8.0, 20 µl of 1 M MgCl₂ and made up to 5 ml with nuclease free water.”

The type of coacervate system used has been specified within each experiment for added clarity:

Page 13, line 321 *“respectively were loaded into bulk CM-Dex : PLys (4:1 final molar ratio) bulk coacervate phase for single turnover conditions i.e. with stoichiometric amounts of ribozyme and substrate, in this case the ribozyme concentration is 2x the concentration of substrate. RNA was extracted from 5 µl of bulk 4:1 CM-Dex : PLys coacervate phase (25 °C) or from 5 µl of 4:1 CM-Dex : PLys coacervate”*

Page 13, line 342 *“into the bulk coacervate phase, CM-Dex : PLys (4:1 final molar ratio)”*

Page 14, line 361 *“**Hammerhead catalysis within coacervate microdroplets.** FRET based HH-min activity within bulk coacervate phase, CM-Dex : PLys (4:1 final molar ratio)”*

Page 15, line 379-380 *“**Determining the diffusion coefficient and the localization of RNA using Fluorescence Recovery after Photobleaching (FRAP)...** within both CM-Dex : PLys and PLys : ATP coacervate microdroplets (4:1 final molar ratio) and the bulk coacervate phase (CM-Dex : PLys 4:1 final molar ratio)”*

Page 16, line 424 *“**Ribozyme activity and localization within coacervate droplets.** To determine the localization of the RNA, CM-Dex: PLys (4:1 final molar ratio) or PLys : ATP (4:1 final molar ratio) coacervate micro-droplets emulsions”*

Page 17, line 444 *“**Fluorescence based partitioning of RNA.** TAM-HH-min, FAM-substrate, FAM-cleaved product, FAM-Tet, FAM-Flex, FAM or TAMRA were loaded into 150 µl dispersions containing 3 µl of bulk coacervate phase (CM-Dex : PLys 4:1 molar ratio) to achieve a final initial concentration of 0.36 µM (cini). After an equilibration time of 10 min, the coacervate phase (3 µl) was separated from the supernatant (147 µl) by centrifugation (10 min at 10000 g). Equal conditions of both fractions were ensured by filling up the 147 µl supernatant fraction to a final volume of 150 µl (Vini) by the addition of 3 µl of RNA-free bulk CM-Dex : PLys (4:1 final molar ratio) coacervate phase (supernatant phase) whilst the 3 µl of bulk 4:1 CM-Dex : PLys (4:1 final molar ratio) coacervate phase fraction was made up to 150 µl (Vini) with RNA-free supernatant (coacervate phase). Coacervates of both samples were dissolved by the addition of 10 µl NaCl (5 M). The Fraction, F, of RNA within either the bulk CM-Dex : PLys (4:1 final molar ratio) coacervate phase or the supernatant phase was obtained using Equation 9.”*

Along this vein, the first instance of “CM-DexLys” should be defined. I was assuming this represents a carboxylated dextran and lysine; if so, the reason for choosing this composition should be given since it is not an obvious prebiotic choice. The composition should be justified in any case.

We thank the reviewer for this comment and have added an additional few sentences to define CM-Dex and PLys (page 2, line 147). For clarity we have included the molecular structures in the SI (new Figure S1). A justification for the composition has also been included in the introduction on page 2, line 46:

“Therefore, coacervate protocells based on Carboxymethyl Dextran sodium salt (CM-Dex) and Poly-L-Lysine (PLys) (Figure S1) were chosen as the model system due to their proven ability to encapsulate and support complex biochemical reactions catalysed by highly evolved enzymes^{9,10}. In contrast to these enzymes, structurally simple ribozymes, which are thought to have played a key role during early biology, lack structural stability and therefore may be rendered inactive by interactions within the highly charged and crowded interior.....”

The additional figure S1, in supplementary information, page 3, line 36

Figure S1: Molecular structures of coacervate components Poly-L-Lysine (PLys), Carboxymethyl-dextran sodium salt (CM-Dex) and Adenosine 5' Triphosphate (ATP) and molecular structures of dye molecules, 5(6)-Carboxytetramethylrhodamine N-succinimidyl ester (TAMRA) and Fluorescein isothiocyanate isomer (FITC/FAM).

Second, the observations of different rate constants and biphasic behavior are presented phenomenologically in the manuscript (results 1-3). I believe more attention, at least discussion, and experiments if possible, regarding possible mechanisms for each of these

observations is needed to understand the potential importance of these results to the field. For example, a statistically significant acceleration of ~5-fold in rate constants was observed in microdroplets vs. bulk phase. The text states a 'slight alternation of the material properties' but the mechanism of such an effect, even if only proposed but not verified, is not obvious to me. Do the authors mean the viscosity? If so, this could probably be measured. A similar concern relates to the biphasic curve (some mention is given of alternate conformers - perhaps this could be tested by chemical probing or even spectroscopically, e.g., CD). An explanation for the quite significant slowing of the ribozyme in the coacervate is also needed given the observation in cited literature of a hammerhead ribozyme accelerated in an ATPS.

We thank the reviewer for this comment and have now included additional discussion regarding the observations of different rate constants. Taking into account the comments from reviewer 1, comment 1, we have also considered that the difference in the rate constants may be attributed to other factors such as diffusion as well as changes in the material properties such as viscosity. To this end we have undertaken additional FRAP experiments on the bulk polymer phase with the expected concentration of FAM-substrate (24.3 μM) within the microdroplet after partitioning and compared the diffusion coefficients and viscosity to a lower concentration of substrate (0.36 μM). The new results show that the viscosity is the same within error. Figure S6 has been modified and replaced with Figure S7 which now shows the diffusion coefficient and viscosities obtained from FRAP of bulk coacervate phase with 0.36 and 24.3 μM of substrate and coacervate microdroplets containing 24.3 μM of substrate. A 50-fold increase in total RNA concentration within the droplets could have an impact on RNA activity due to variations in water activity (Nakano et al., JACS (2009) 131: 16881) or the dielectric constant of the environment (Nakano et al., FEBS Open Bio (2014) 4: 643).

We have modified and added the following text to the main text (page 5, from line 222):

.... "From the partition coefficient we calculated concentrations of 49.6 μM HH-min and 24.3 μM substrate in a single microdroplet compared to 1 μM HH-min and 0.5 μM substrate within the bulk coacervate phase. Whilst the observed rate differences between the bulk coacervate and coacervate microdroplet phases could be due to variations in viscosity, quantitative FRAP analysis with two different FAM-substrate concentrations (0.36 μM and 24.3 μM) showed that the measured viscosities of the bulk coacervate phase and coacervate microdroplet are comparable within error (Figure S7). Therefore, secondary effects arising from the increased RNA concentration within the coacervate microdroplet phase may be responsible for the increased rate constants observed. The ribozyme and substrate concentrations are approximately 50 times more in the coacervate microdroplet (49.6 μM and 24.3 μM respectively) compared to the bulk coacervate phase (1 μM and 0.5 μM respectively). The difference in diffusion length scales of RNA in the microdroplet environment could lead to increased saturation of the ribozyme and therefore greater apparent rate constants in the coacervate microdroplets compared to bulk coacervate phase. In addition, enrichment of RNA could lead to changes in material properties such as water activity or dielectric constant, which have a direct impact on the rate of hammerhead-catalysed RNA cleavage.^{35,36} Thus, secondary effects that result from the increased RNA concentration within coacervate microdroplets may concomitantly contribute to an increase in the apparent rate constant."

Additional references have been added to page 24, line 591-596

35. Nakano, S., Karimata, H. T., Kitagawa, Y. & Sugimoto, N. Facilitation of RNA Enzyme Activity in the Molecular Crowding Media of Cosolutes. *J. Am. Chem. Soc.* **131**, 16881–16888 (2009).

36. Nakano, S.-I., Kitagawa, Y., Miyoshi, D. & Sugimoto, N. Hammerhead ribozyme activity and oligonucleotide duplex stability in mixed solutions of water and organic compounds. *FEBS Open Bio* **4**, 643–650 (2014).

A new Figure S7 with a new figure legend has been placed supplementary information on page 9.

Figure S7: FRAP of bulk coacervate phase (CM-Dex : PLys (4:1 final molar ratio)) and coacervate microdroplets (CM-Dex : PLys (4:1 molar ratio)) containing FAM-substrate **(A)** Diffusion coefficients (blue) (D) and viscosities (orange) (η) for (i) – 0.36 μM FAM-substrate in bulk coacervate phase, $D = 1.56 \pm 0.11 \mu\text{m}^2/\text{s}$, $\eta = 109 \pm 8 \text{ mPa}\cdot\text{s}$ ($n = 14$), (ii) – 24.3 μM FAM-substrate, which is the calculated concentration in microdroplets after accumulation based on the partitioning experiment (Figure S6), in bulk coacervate phase, $D = 1.35 \pm 0.13 \mu\text{m}^2/\text{s}$, $\eta = 127 \pm 12 \text{ mPa}\cdot\text{s}$ ($n = 10$) and (iii) – 24.3 μM FAM-substrate (overall concentration of 0.36 μM) within microdroplets $D = 1.76 \pm 0.42 \mu\text{m}^2/\text{s}$, $\eta = 102 \pm 28 \text{ mPa}\cdot\text{s}$ ($n = 18$). **(B)** Output frames of (ii) (scale bar 5 μm) from confocal imaging (63x) are shown at $t = -1$ s before bleaching, directly after bleaching (red circle, $t = 0$ s) and $t = 29$ s after bleaching. The fluorescence intensity was normalized against a reference (green circle). **(C)** Plots of normalised FRAP data for (ii) show the data distribution and averaged fit from fitting to Equation 5 (blue).

Modifications have been made to materials and methods to reflect the new experiments, page 15, lines 379-382.

...“undertaken within both CM-Dex : PLys and PLys : ATP coacervate microdroplets (4:1 final molar ratio) and the bulk coacervate phase (CM-Dex : PLys 4:1 final molar ratio) containing either TAM-HH-min (0.36 μM), FAM-substrate (0.36 μM), FAM-substrate (24.3 μM), FAM-cleaved substrate (0.36 μM).”

With regard to the change in the transition from the mono-exponential to biphasic behavior within the coacervate phase compared to the buffer solution. We have undertaken CD spectra, which show a decrease in the secondary structure and a small shift in the peak maxima. This supports the hypothesis that there are heterogeneous ribozyme populations with different secondary structure, conformational and equilibrium states. Therefore, we have changed the text accordingly (page 4, lines 90-97).

From:

“This may be attributable to heterogeneous ribozyme populations with alternative conformational and equilibrium states, as observed for some HH systems in aqueous buffer conditions.^{31,32} It is possible that the charged and crowded coacervate microenvironment affects the structure of HH-min, restricting substrate binding, sterically hindering substrate-enzyme complex formation and/or spatially restricting diffusion of the cleavage assay components. “

To:

“ This may be attributable to heterogeneous ribozyme populations with variations to secondary structure and / or alternative conformational and equilibrium states, as observed for some HH systems in aqueous buffer conditions.^{31,32} Circular dichroism (CD) spectra show a reduction in Molar Ellipticity ($[\theta]$) for HH-mut within bulk coacervate phase compared to aqueous buffer with a small commensurate shift in the peak maxima from 265 nm to 268 nm respectively (Figure S4). These results show that the fold of HH-mut is altered in the polyelectrolyte- rich bulk coacervate phase, with an overall loss of secondary structure that could affect catalytic activity. ”

A new figure showing the CD-data has been placed in the SI as Figure S4.

Figure S4: Circular Dichroism Spectroscopy of 2 μ M HH-mut in cleavage buffer (orange line), 1 μ M HH-mut in cleavage buffer (yellow line) and 2 μ M HH-mut in CM-Dex : PLys, 4:1 final molar ratio bulk coacervate phase (blue line) shown between 250 nm and 310 nm to focus on the RNA secondary structure [1]. The shaded region shows the standard deviation from 3 spectra, each an average of 5 repeats (orange and yellow) and 13 spectra, each an average of 10 repeats (blue). All spectra are shown with the background (either cleavage buffer or bulk polymer phase) removed.

Additional protocols for the materials and methods have been described on page 19, line 486-495:

“Circular Dichroism. To investigate the secondary structure of the hammerhead ribozyme, HH-mut was mixed into either buffer ($c_{HH} = 1$ or 2μ M) or bulk coacervate phase ($c_{HH} = 2 \mu$ M) (CM-Dex : PLys, 4:1 molar ratio) and loaded into a 1 mm Special Quartz Cuvette (200 μ L). CD spectra were measured using a ChirascanTM-Plus CD Spectrometer (Applied Photophysics), with data collected every 1 s at 25 °C from 320 to 200 nm with a resolution of 1 nm. Either 5 or 10 repeat spectra were measured for buffer and bulk coacervate samples respectively. Background spectra of buffer alone and bulk coacervate phase were taken under the same conditions and were subtracted from the spectra of HH-mut in buffer and bulk coacervate phase respectively. All spectra were offset at 320 nm, normalised for the cuvette pathlength and converted from ΔA to Molar Ellipticity ($[\theta]$) (deg · cm² · dmol⁻¹). ”

Some additional comments are given below.

Pg 2: What exactly does kick starting mean in this context?

Kickstarting means to get going i.e. the same as initiating. The text has therefore been changed from (page 2, line 28):

“by kick-starting and concentrating”

To:

“concentrating molecules and thus initiating the first biochemical reactions on Earth”

Page 3: I am not sure if this 60-fold (?) difference should be called ‘an order of magnitude’; it would be better to simply give the fold-difference.

This has been changed from page 4, line 86:

“Thus, the fastest rate constant k_1 is an order of magnitude slower than in buffer conditions ($k_0 = 0.6 \pm 0.2 \text{ min}^{-1}$)”

To:

“Thus, the fastest rate constant k_1 is 60 fold slower than in buffer conditions ($k_0 = 0.6 \pm 0.1 \text{ min}^{-1}$)”

Figure 1C: The curve fit equations should be given explicitly with coefficients and constants. Does the curve begin from 0? If not, why not? What is the dead time of this experiment? Is (i) showing a different experiment from (ii); if so, how are these different? How do the grey points in (ii) indicate standard deviation - why would the stdev increase over time?

The equations for the fitting are in the materials and methods. It is not common practice to place the equations into the figure legends, therefore, the references have been placed into the figure legend to describe where the equation can be found in the materials and methods (see below (a)). The dead time or the offset time and in figure 1c and has been described in the materials and methods (see below (b)). Additionally, the curve fit equations have been given explicitly with the coefficients and constants (see below, (c)). i and ii show the same experiment and this is described in a new figure legend (see below, (a)), (i) is a fit and its residual to a single experiment and (ii) shows the mean (blue line) and standard deviation (grey data) from at least 5 experiments. The difference in standard deviation over time is due to difference in variance between the five experiments. For clarity the figure legend has been modified (see below, (a)).

(a) Figure legend 1C has been modified on page 26, line 628 from...

“(C) Real-time cleavage kinetics in 10 mM Tris•HCl pH 8.3 and 4 mM MgCl₂. (i) A monoexponential fit (grey line) to kinetic data (grey dots) and residuals of the fit (inset) (ii) Blue line is the mean of the individual fits (blue line). Grey data points represents the standard deviation ($N \geq 5$) from experimental data. “

To:

“(C) (i) A monoexponential fit (materials and methods, Equation 3) (grey line) to kinetic data (grey dots) and residuals of the fit (inset) (ii) mean of the individual fits to each experiment (Blue line) with the standard deviation of the mean of the fits (grey data points) ($N = 5$). “

In the materials and methods, all the terms have additionally described:

(b) page 18, line 356

“ t_0 is the dead time between sample preparation and the first measurement (separately measured)”

(c) page 11 line 308:

“Kinetic profiles were fit to either single exponential growth (Equation 1) under buffer conditions or bi-exponential growth (equation 2) for bulk coacervate experiments.

$$I(t) = 1 - e^{-\left(\frac{t-t_0}{\tau}\right)} \quad (\text{Equation 1})$$

$$I(t) = 1 - (A_1 e^{-\left(\frac{t-t_0}{\tau_1}\right)} + (1 - A_1) e^{-\left(\frac{t-t_0}{\tau_2}\right)}) \quad (\text{Equation 2})$$

The corresponding rate constants k_1 or k_2 are obtained from the fitted time constants τ_1 or τ_2 where $k=1/\tau$.”

To:

“ The first order time constant τ was obtained by fitting the cleaved and uncleaved kinetic profiles to single exponential fits (Equations 1 and 2 respectively) and was optimised globally.

$$I(t)^{\text{cleaved}} = I_{\text{max}}^{\text{cleaved}} - I_{\text{max}}^{\text{cleaved}} \cdot e^{-\left(\frac{t}{\tau}\right)} + C \quad (\text{Equation 1})$$

$$I(t)^{\text{uncleaved}} = C + (I_{\text{max}}^{\text{uncleaved}} - C) \cdot e^{-\left(\frac{t}{\tau}\right)} \quad (\text{Equation 2})$$

$I(t)$ is the band intensity, C is an offset and τ the first order time constant. The first order rate constant (k_0) is then determined by $1/\tau$.”

And on page 14, line 355-358

Where $I(t)$ is the normalized intensity, A_1 is the amplitude of the first $I - A_1$ the amplitude of the second exponential, t is the time in min, t_0 is the dead time between sample preparation and the first measurement (separately measured), τ_1 or τ_2 are the fitted time constants. The corresponding rate constants k_1 or k_2 are obtained from τ_1 or τ_2 where $k=1/\tau$.”

Figure 1D: Again, the curve fit equation should be given. These curves appear to begin at 0, but the relative amplitude of the two exponential fits should be given. Also, a supplementary figure should be included to show this graph as a semilog plot to demonstrate the two regimes. Same question as Fig. 1C with respect to stdev.

As discussed previously, it is not usual practice to place the fit equations within the figure legends, therefore, the curve fit equation has been referred to within the figure legend (see below). In addition, to clarify the meaning of the standard deviations, the figure legend has been rewritten: Page 26, line 632.

From:

“ (i) Biexponential fit (materials and methods Equation 2) (dark grey line) to experimental data (grey dots) with the residuals (inset) (ii) mean biexponential fit (orange) of individual fits ($N \geq 5$). Grey data points represent the standard deviation ($N \geq 5$) from the experimental data.”

To:

“(i) Cleavage in bulk coacervate phase (normalized to the amount of cleaved product at $t = 530$ min from gel electrophoresis). (i) Biexponential fit (materials and methods, Equation 4) (dark grey line) to experimental data (grey dots) with the residuals (inset) (ii) mean biexponential fit (orange) of individual fits ($N \geq 5$). Grey data points represent the standard deviation ($N = 5$) from the experimental data.”

The amplitudes of the two exponential fits have been added to SI table 3:

	k_0 (min ⁻¹)	k_1 (min ⁻¹)	k_2 (min ⁻¹)	A_1	A_2
Buffer	0.6 +/- 0.1			0.94 +/- 0.02	
Bulk coacervate phase		$1.0 \times 10^{-2} \pm 0.1 \times 10^{-2}$	$1.5 \times 10^{-4} \pm 0.8 \times 10^{-4}$	0.23 +/- 0.03	0.77 +/- 0.03
Coacervate microdroplets		$4.4 \times 10^{-2} \pm 1.3 \times 10^{-2}$	$2.3 \times 10^{-3} \pm 0.2 \times 10^{-3}$	0.31 +/- 0.20	0.69 +/- 0.20

For repeat experiments, the relative amplitudes have been described in the relevant figure legend of figure S5 (supplementary information, page 7 line 100).

“Figure S5: Reproducibility of HH-min – FRET-substrate assay: The FRET assay was repeated with a different batch of HH-min and FRET-substrate (final concentrations of 1 μM for HH-min or HH-mut and 0.5 μM FRET-substrate) in cleavage buffer, bulk coacervate phase or coacervate microdroplets. Time-resolved kinetics was measured in (A) cleavage buffer (10 mM Tris•HCl pH 8 and 4 mM MgCl₂) or (B) bulk coacervate phase (CM-Dex : PLys, 4:1 final molar ratio) using a plate reader or (C) in microdroplets (CM-Dex : PLys, 4:1 final molar ratio) by fluorescence optical microscopy at final concentrations of 1 μM for HH-min and 0.5 μM FRET-substrate. Time traces of (A) could be fit with a single exponential (Equation 3) ($k_0 = 0.5 \pm 0.1 \text{ min}^{-1}$). Time traces for (B) and (C) were fit with double exponentials (Equation 4) yielding rate constants and amplitudes of k_1 of $5 \times 10^{-3} \pm 2 \times 10^{-3} \text{ min}^{-1}$ ($A_1 = 0.31 \pm 0.05$) and k_2 of $2.9 \times 10^{-4} \pm 0.2 \times 10^{-4} \text{ min}^{-1}$ ($A_2 = 0.69 \pm 0.05$) for (B) and k_1 of $3.0 \times 10^{-2} \pm 0.1 \times 10^{-2} \text{ min}^{-1}$ ($A_1 = 0.24 \pm 0.04$) and k_2 of $3.0 \times 10^{-3} \pm 0.3 \times 10^{-3} \text{ min}^{-1}$ ($A_2 = 0.76 \pm 0.03$) for (C).”

As suggested by the reviewer we have produced a semi log plot of the biexponential phase to show the two regimes as shown below of the data shown in Figure 3b:

Figure: RNA catalysis in coacervate microdroplets. Semi Log plot of Background corrected and volume / endpoint normalized fluorescence intensity of droplets.

However, this plot does not add any additional information to what has already been shown in figures S3 (was figure S2) and figure 3 where we show single and bi-exponential fits to the cleaved fraction as a function of time in the bulk coacervate phase and in the coacervate microdroplets and the corresponding residuals (see below).

Figure S3: Kinetic analysis for HH-min FRET assay in (A) bulk coacervate phase (CM-Dex: PLys, 4:1 final molar ratio) (A) or (B) a dispersion of coacervate microdroplets (CM-Dex:PLys, 4:1 final molar ratio). The assay was undertaken in limited substrate conditions at total concentrations of $1 \mu\text{M}$ for HH-min and $0.5 \mu\text{M}$ FRET-substrate. Time resolved data was either from (A) a plate reader experiments or (B) optical microscopy. The increase in fluorescence intensity from cleaved product (grey data) was plotted as a function of time and the thickness of the grey shaded region is the standard deviation from 5 repeats (A) and 12 individual droplets (B). Single exponential fits to the data show non-random residuals (inset) indicating that fits to single-exponential decays are not reliable.

Figure 3. RNA catalysis in coacervate microdroplets. (A) (i) Optical microscopy images of CM-Dex : PLys (4:1 final molar ratio) coacervate microdroplets prepared in cleavage buffer ($1 \mu\text{M}$ of HH-min and $0.5 \mu\text{M}$ FRET-substrate). Fluorescence microscopy images at $t = 0 \text{ min}$ (ii) and $t = 900 \text{ min}$ show an increase in FAM fluorescence (see inset). Scale bars are $20 \mu\text{m}$. (B) Background corrected and volume / endpoint normalized fluorescence intensity of droplets. Standard deviation of kinetics from twelve micro-droplets (grey) with the mean biexponential fit (blue) and residuals (inset).

Figure 3A: Axes labels on insets are needed. I assume these are distributions of intensity; some comment on the procedure for intensity normalization between samples is needed.

We thank the reviewer for this suggestion and have now added the axes labels into the plot. The new figure is shown below and is found in the main text on page 27.

Figure 3. RNA catalysis in coacervate microdroplets. (A) (i) Optical microscopy images of CM-Dex : PLys (4:1 final molar ratio) coacervate microdroplets prepared in cleavage buffer (1 μ M of HH-min and 0.5 μ M FRET-substrate). Fluorescence microscopy images at $t=0$ min (ii) and $t=900$ min show an increase in FAM fluorescence (see inset). Scale bars are 20 μ m. (B) Background corrected and volume / endpoint normalized fluorescence intensity of droplets. Standard deviation of kinetics from twelve micro-droplets (grey) with the mean biexponential fit (blue) and residuals (inset).

For this experiment, the intensities were normalized to the amount of cleaved product, which was obtained from gel electrophoresis. This is explained in the materials and methods via the following text in the materials and methods on page 13, line 333:

“Band intensities were measured using ImageQuant at a specific time point and uncleaved substrate was corrected by the FRET effect factor (1.69) (Figure S12). The fraction of cleaved substrate of the total sum of cleaved and uncleaved substrate was determined and used to normalize kinetic data obtained from spectroscopy or microscopy.”

And (page 15, line 372):

“The amount of substrate cleaved was normalised by gel electrophoresis at a given time point as described previously. Kinetic parameters were derived from fitting the kinetic data to Equation 4. To test for reproducibility the experiment was repeated with a different batch of HH-min using the same experimental conditions (Figure S5).”

Why would FRAP recovery only reach 70% for HH-min? This needs at least a proposed explanation.

Yes, we agree with the reviewer that this is an interesting observation. As such, there is already a proposed explanation in the main text on page 6, line 146-148:

“In comparison, TAM-HH-min showed only 70 % recovery after 300 s with $\tau = 189 \pm 14$ s, attributed to either a low concentration of HH-min within the surrounding aqueous phase and/or a slow rate of transfer into the coacervate droplet from its exterior (Figure S6).”

Semantic issue: I do not think ‘protocell’ is normally used to describe coacervates; this could potentially be confusing to readers who understand the word ‘protocell’ to be a membrane-bound compartment. I would recommend eliminating this word from the title.

The protocell is a broad term which describes any compartment which could be a precursor to the first cells as e.g. described by the Walde¹ and Chen², and Szostak³ labs. A range of different

membrane bounds compartments and membrane free compartments (including coacervates) have been called protocells^{4,5}. In fact, the name protocells was already used for membrane-free compartments in the 1970s⁶. Thus, membrane-bound compartments are just one class of compartments that fall under the umbrella term of protocells.

1. Walde, P. Building artificial cells and protocell models: experimental approaches with lipid vesicles. *Bioessays* 32, 296–303 (2010).
2. Saha, R. & Chen, I. A. Origin of Life: Protocells Red in Tooth and Claw. *Curr Biol* 25, R1175-1177 (2015).
3. Jia, T. Z., Hentrich, C. & Szostak, J. W. Rapid RNA Exchange in Aqueous Two-Phase System and Coacervate Droplets. *Orig Life Evol Biosph* 44, 1–12 (2014).
4. Huang, X. *et al.* Interfacial assembly of protein–polymer nano-conjugates into stimulus-responsive biomimetic protocells. *Nat Commun* 4, 2239 (2013).
5. Yin, Y. *et al.* Non-equilibrium behaviour in coacervate-based protocells under electric-field-induced excitation. *Nat Commun* 7, 10658 (2016).
6. Snyder, W. D. & Fox, S. W. A model for the origin of stable protocells in a primitive alkaline ocean. *BioSystems* 7, 222–229 (1975).

Therefore, we have kept the term protocell in the title and do not think it would be confusing for readers.

Reviewer #3 (Remarks to the Author):

The study entitled “Compartmentalized RNA catalysis in membrane – free coacervate protocells” reported the ribozyme activity in coacervate and size-dependent diffusion of RNAs among coacervate microdroplets. The authors analyzed the kinetics of a ribozyme and diffusion rate in detail. The manuscript was clearly written and the measurement and analysis were well performed. However, I have two major concerns as described below and they are critically important for the significance of this study to readers in a broad field of science. Therefore, I cannot recommend this manuscript for publication in Nature Communications.

Major points

1. The first concern is about novelty. I agree that ribozyme activity has not been observed in coacervate. But there have been several studies about other biochemical reactions, such as actinorhodin polyketide synthesis and gene expression, as the authors cited in references 8-9. Therefore, ribozyme reaction in coacervate is not surprising and I don't believe this is a significant advance. I admit that the authors precisely analyzed the kinetics and diffusion and this study is worth publishing, but a more specific journal would be suitable.

The reviewer is correct and there have been previous studies, which have demonstrated enzymatic activities within the coacervate. However, the enzymes involved in these studies are highly evolved, large and structurally complex macromolecules and not primitive structures like the minimal form of our 39 nt hammerhead ribozyme. The ribozyme lacks the structural integrity of a protein enzyme and is therefore much more susceptible to environmental conditions. In particular, the crowded polyelectrolyte environment of the coacervate bulk phase could affect the secondary structure and therefore, it is far from obvious that the ribozyme would retain its activity under these conditions. Despite this, our studies show for the first time that primitive RNA catalysis within coacervate microdroplets. We believe that our results are novel and interesting for a broad audience including the Origins of life, synthetic biology and the biology communities.

An additional sentence has been added to the main text to explain why the results that we show cannot be assumed and therefore demonstrates novelty (page 2, from line 46).

“Therefore, coacervate protocells based on Carboxymethyl Dextran sodium salt (CM-Dex) and Poly-L-Lysine (PLys) (Figure S1) were chosen as the model system due to their proven ability to encapsulate and support complex biochemical reactions catalysed by highly evolved enzymes^{9,10}. In contrast to these enzymes, structurally simple ribozymes, which are thought to have played a key role during early biology, lack structural stability and therefore may be rendered inactive by interactions within the highly charged and crowded interior. Herein, we directly probe the effect of the coacervate microenvironment on primitive RNA catalysis, and show the ability of the coacervate microenvironment to support RNA catalysis whilst selectively sequestering ribozymes and permitting transfer of lower molecular weight oligonucleotides.”.

2. The second concern is about one of the main claim of this study, selective retention of a long RNA in coacervate. I understood that this claim based on the result of Figure 4. I think there are two problems in this experiment to withdraw the conclusion.

First, the localization of the substrate RNA was not directly measured. I agreed that the HH-RNAs were localized in region 1, whereas the localization of the substrate RNA was not directly determined. Instead, the authors observed the FRET signal produced by ribozyme activity, which depends on the localization of both ribozyme and substrate and thus cannot be regarded as a direct evidence of substrate localization. Direct measurement of the substrate localization should be measured to conclude the different retention time.

We thank the reviewer for this comment and agree that figure 4 shows localisation of the ribozyme and substrate. However, we also base our conclusions regarding length dependent sequestration in the coacervate phase by direct measurement of the substrate as discussed by the reviewer. This is shown in the supplementary information (previous Figure S7, now Figure S10, supplementary figure page 12), which describes localisation experiments where only the TAM-HH-min or FAM-substrate were encapsulated into coacervate microdroplets and loaded into one end of the channel with coacervate droplets containing non-fluorescent HH-mut were loaded into the other end of the channel.

Figure S10: Localization experiments - Controls: (A) CM-Dex : PLys Coacervate microdroplets (4:1 final molar ratio) containing either 0.36 μM of TAM-HH-min (i) or 0.36 μM FAM-substrate (ii) was loaded into a microscope channel and CM-Dex : PLys coacervate microdroplets (4:1 molar ratio) containing 0.36 μM HH-mut was loaded into the other end of the channel. Optical microscopy images were taken every 5 min for 90 min with $\lambda_{\text{exc}} = 542 \pm 13.5$ nm and $\lambda_{\text{em}} = 593 \pm 23$ nm and in the FAM channel with $\lambda_{\text{exc}} = 469 \pm 20$ nm and $\lambda_{\text{em}} = 525.5 \pm 23.5$ nm for TAM-HH-min and FAM-substrate respectively. (B) Optical microscopy images from position 3 shows no increase in fluorescence intensity after 90 min whilst optical microscopy images show an increase in fluorescence intensity of FAM-substrate in position 1 over time. (C) plots of the mean integrated fluorescence intensities (solid line) and the standard deviation (shaded area) of at least 7 droplets show no diffusion of TAM HH-min through the length of the channel (i) whilst the increase in fluorescence intensity in region 2 shows diffusion of the FAM-substrate from region 1.

For more clarity, an additional sentence has been added into the main text (page 7, line 176-178) to refer to these localisation experiments:

“Control experiments probing the transfer of FAM-substrate only, show increased fluorescence intensity in regions 2 and 3 giving a direct confirmation that the 12-mer substrate diffuses between droplets (Figure S10).”

Second, the difference in the retention of the two RNAs can be attributed to other factors than RNA sizes, such as labeled fluorescent compound (TAM or FAM), RNA sequences, or RNA secondary structures. To verify the effect of RNA size, additional experiments with various size and sequence of RNAs are required.

Yes, we agree with the reviewer and therefore have undertaken additional partition experiments with 6 mer RNA (cleaved product), TAMRA, FAM and two additional 12 mer RNAs which have different sequences (FAM-Tet and FAM-Flex). The latter two sequences for RNAs do not fold in whilst the FAM-substrate (12-mer) will fold on itself. Our results show that the partitioning of FAM and TAMRA are an order of magnitude lower than the TAMRA-HH-min, FAM-substrate and FAM-cleaved product which indicates that the partitioning in these RNAs used for the HH-ribozyme assay is driven by the RNA and not the dye molecule that is attached to it. In addition, we show that partitioning of the RNA *is* sequence dependent as our results show an order of magnitude difference in the partition coefficient for FAM-Tet and FAM-Flex compared to the pyrimidine-rich FAM-substrate (all 12mers).

With the additional experiments, additional sentences have been added to the main text to describe the trends, new figures have been produced with the additional partition coefficients and placed into the SI as Figure S6 and S9. A description of the trends are included in the figure captions and additions have been included into the materials and methods:

New text has been added to the main text at page 6, line 153:

“Additionally, we find that the RNA sequence can have an impact on partitioning where an increase in purine content decreases partitioning of 12mer RNAs approximately 10-fold compared to the highly pyrimidine-rich HH-substrate (Figure S9). For the RNAs specific to our HH ribozyme assay, the general selective retention of longer length polynucleotides with transfer of shorter length RNA can have interesting implications for ribozyme catalysis within coacervate droplets.”

And on Page 8, line 207.

“Moreover, our results also show that the strength of oligonucleotide selectivity is dependent on the composition of the coacervate microdroplets and the molecular sequence of RNA.”

Additional SI figures describing the new partition experiments in supplementary information page 8.

Figure S6: Partitioning of three different length RNA (i) 39-mer TAM-HH-min, (ii) 12-mer FAM-substrate, (iii) 6-mer FAM-Cleaved-substrate and (iv) FITC (v) TAMRA in the 4:1 (final molar ratio) CM-Dex : PLys coacervate system (A-C). To obtain the partition coefficient 0.36 μ M (total concentration) of either dye tagged RNA or dye, each sample was incubated in 150 μ L CM-Dex / PLys coacervate microdroplet solution (3 μ L bulk coacervate phase + 147 μ L supernatant) and left to equilibrate for 10 min. The dispersion was then centrifuged and the supernatant separated from the coacervate phase. Either 3 μ L of the bulk coacervate phase or 147 μ L supernatant was made up to 150 μ L by the addition of 147 μ L RNA free supernatant or 3 μ L RNA free coacervate, respectively. The fractions of dye tagged RNA or dye in the coacervate phase (A) and in the supernatant phase (B) were obtained spectroscopically. Partitioning coefficients were calculated using Equation 7-9 and plot for the CM-Dex / PLys coacervate system (C) with the y-axis plot on the logarithmic scale. The results show the general trend that longer length RNA partitions more into the coacervate phase compared to shorter length RNA with (i) 39-mer TAM-HH-min ($K = 9600 \pm 5600$), (ii) 12-mer FAM-substrate ($K = 3000 \pm 2000$), (iii) 6-mer FAM-Cleaved-substrate ($K = 1500 \pm 300$). The partition coefficients for (iv) FITC ($K = 160 \pm 14$), (v) TAMRA ($K = 83 \pm 10$) are an order of magnitude less than the dye tagged RNA showing that partitioning is primarily driven by the nucleotide and not the dye molecule.

Please note that the numbers for the partition coefficients for the 39-mer TAM-HH-min, 12 mer FAM substrate and the 6-mer FAM-cleaved substrate are different to what we previously submitted. This is because, additional experiments were undertaken and the new partition coefficients are the calculated averages and standard deviations from all of the experiments.

In supplementary information on Page 9:

Figure S9: Partitioning (K) of three different sequences RNA (i) 12-mer FAM-substrate (ii) 12-mer FAM-Flex (iii) 12-mer FAM-Tet. To obtain the partition coefficient 0.36 μ M (total concentration) of FAM tagged RNA was incubated in 150 μ L CM-Dex / PLys coacervate microdroplet solution (3 μ L bulk coacervate phase + 147 μ L supernatant) and left to equilibrate for 10 min. The dispersion was centrifuged and the supernatant phase separated from the coacervate phase. Either 3 μ L of the bulk coacervate phase or 147 μ L supernatant was made up to 150 μ L by the addition of 147 μ L RNA free supernatant or 3 μ L RNA free coacervate, respectively. The fractions of FAM tagged RNA in the coacervate phase and in the supernatant phase were obtained spectroscopically and the partitioning coefficients were calculated using Equation 7-9 and plotted with the y-axis on the logarithmic scale. Corresponding partitioning coefficients are (i) 12-mer FAM-substrate ($K = 3000 \pm 2000$, $n = 20$), (ii) 12-

mer FAM-Flex (K = 114 +/- 33, n = 9), (iii) 12-mer FAM-Tet (K= 126 +/- 10, n = 9). The results show that the partitioning of RNA is not only dependent on its length but also on its sequence/structure.

**Additions have been made the materials and methods due to the additional experiments:
To the materials on page 9, line 236.**

“Fluorescein isothiocyanate isomer I (FITC, C₂₁H₁₁NO₅S, 389.38 g/mol)”

Page 9, line 244-246.

..... 5(6)-Carboxytetramethylrhodamine succinimidyl ester (TAMRA, C₂₉H₂₅N₃O₇, 527.53 g/mol) was purchased from Thermo Fisher Scientific.”

Page 17, line 442:

“Fluorescence based partitioning of RNA. TAM-HH-min, FAM-substrate, FAM-cleaved product, FAM-Tet, FAM-Flex, FITC and TAMRA”

Additions to the Supplementary information describing the new RNA sequences on Page 2, Table S2:

Name	5' tag	Sequence 5' -> 3'	3' tag	Length (nt)	Mw (kDa)	λ _{exc} (nm)	
FAM-Flex	FAM	GCAGAACGAAGC		12	4.6	495	
FAM-Tet	FAM	GCACUUCGGUGC		12	4.5	495	

Minor point:

The subsection of “Preparation of bulk coacervate phase...” of Materials and Method section, was difficult to understand. I believe the problems is lack of explanation of several phases: coacervate phase, polymer phase, bulk polymer phase, supernatant phase. Is the “coacervate phase” same as polymer phase? Clarification of these terms would be kind for readers.

Add in explanation into the materials and methods of what these phases are.

We thank the reviewer for the comment and have included an additional explanation of the coacervate phase and the microdroplet in the materials and methods (page 11, line 279).

“Preparation of bulk coacervate phase and coacervate microdroplets containing RNA HH and substrate. Preparation of bulk coacervate phase and coacervate microdroplets containing RNA HH and substrate. Aqueous dispersions of CM-Dex:PLys coacervate microdroplets (4:1 molar ratio) or ATP : PLys (4:1 molar ratio) in 10 mM Tris•HCl pH 8.0 and 4 mM MgCl₂ were first prepared by mixing 200 μl of 1 M CM-Dex, 250 μl of 0.2 M PLys, 50 μl of 1 M Tris•HCl pH 8.0, 20 μl of 1 M MgCl₂ and made up to 5 ml with nuclease free water. Aqueous dispersions of PLys:ATP coacervate dispersions (4:1 molar ratio) in 10 mM Tris•HCl pH 8.0 and 4 mM MgCl₂ were produced by mixing 1000 μl of 0.2 M PLys, 500 μl of 0.1 M ATP, 50 μl of 1 M Tris•HCl pH 8.0, 20 μl of 1 M MgCl₂ and made up to 5 ml with nuclease free water. To

produce a polymer only bulk coacervate phase, the aqueous dispersion of microdroplets was centrifuged (10 min, 4000 g) and the supernatant removed. To produce ribozyme (final concentration: 1 μ M) and substrate (final concentration: 0.5 μ M) loaded samples, RNA was then mixed with 1 μ l of the bulk coacervate phase, followed by either centrifugation (10 min, 4000 g) of the mixture and removal of supernatant to produce a loaded bulk phase (CM-Dex / PLys only) or vortexing of the mixture with 49 μ l supernatant to produce a dispersion of RNA loaded coacervate microdroplets.”

This has been referred to in the main text when the bulk coacervate phase and the microdroplets are first mentioned (page 3, line 62):

“incubated within a bulk polysaccharide / polypeptide coacervate phase or within coacervate microdroplets (see materials and methods)”

Additionally, all of the terminology has been changed to maintain consistency i.e. everything is termed either bulk coacervate phase to describe both polymer phase and bulk polymer phase and the supernatant phase to describe the polymer poor phase, for example, in the materials and methods, page 17, lines 446-452:

“To ensure both fractions were treated equally the 147 μ l supernatant fraction was filled to a final volume of 150 μ l (V_{ini}) by the addition of 3 μ l of RNA-free bulk CM-Dex : PLys (4:1 final molar ratio) coacervate phase (supernatant phase), whilst the 3 μ l of RNA-containing bulk 4:1 CM-Dex : PLys (4:1 final molar ratio) coacervate phase fraction was made up to 150 μ l (V_{ini}) with RNA-free supernatant (coacervate phase). Coacervates of both samples were dissolved by the addition of 10 μ l NaCl (5 M). The Fraction, F, of RNA within either the bulk CM-Dex : PLys (4:1 final molar ratio) coacervate phase or the supernatant phase was obtained using Equation 9.”

Reviewers' comments:

Reviewer #1 (Remarks to the Author):

I appreciate the obvious effort and attention to detail that went into the revised manuscript. My only remaining question relates to the observed differences between ribozyme concentration, and thus activity in bulk coacervates vs. a microdroplet dispersion. My sense is that the authors have tried to lay out a possible explanation as to why these differences have been observed. However, a lack of clarity in methodology and the way in which these explanations are presented seem to suggest an interesting observation, rather than a potential flaw in the experiment.

My confusion begins on page 4, where the authors state, "The final concentration of enzyme and substrate in the microdroplet dispersion was equivalent to the final concentration of the bulk coacervate phase under single turnover conditions (1 μM of HH-min and 0.5 μM FRET-substrate)." However, on the following page, they claim that the concentrations in the two phases are in fact different: "From the partition coefficient we calculated concentrations of 49.6 μM HH-min and 24.3 μM substrate in a single microdroplet compared to 1 μM HH-min and 0.5 μM substrate within the bulk coacervate phase." How is this possible? Coacervates are known to be equilibrium phases. Therefore, whether the material is present as a droplet or as a larger bulk phase, it should not be expected to have a different concentration.

Related to this, I was unclear as to how the samples were prepared. In the manuscript, the text suggests that the droplet dispersions were created by resuspending a bulk coacervate. However, there is no discussion of how this resuspension was achieved in the methods section for the activity experiments. Instead, there is a description of how an RNA-containing solution was mixed with either bulk coacervate or a dispersion of droplets. While care was taken to assure that the total volume of coacervate was the same between these two samples, there is no discussion about potential differences in equilibration time that would be critical based on the different geometries (and thus diffusion distances) between these two samples. I am very concerned that the observation of higher concentration and thus activity in the microdroplets is due to differences in the ability of the two samples to equilibrate because of differences in geometry, rather than an intrinsic difference between the two states of the material.

In contrast, the partitioning experiments allow equilibration of RNA with a suspension of microdroplets for 10 minutes, followed by centrifugation to form a bulk coacervate phase that is assayed. Based on this procedure, I do not see how there could be a difference in the partitioning between droplets and bulk phase as both materials are used in the assay.

The only possible way in which I could envision a difference in RNA concentration between bulk and droplet materials would be if the RNA selectively binds to the droplet surface. I was unsure whether the fluorescence images shown in Figures 3 and 4 were confocal or wide-field. However, confocal imaging would clearly highlight if there were such a concentration of RNA at the surface. Additionally, I would be surprised if the concentration of the droplet could be increased by 50x in such a manner.

The authors must clarify this point in the text. Additionally, it would be possible to test for differences in equilibration by assaying the concentration of RNA in the supernatant above a bulk coacervate material as a function of time.

Minor Comments:

1. Please specify in the caption whether the fluorescence micrographs in Figures 3 and 4 from

confocal or wide-field imaging.

2. In the methods section on page 17 related to fluorescence partitioning experiments, the text reads "to achieve a final initial concentration of 0.5 μM ." I believe that "final" should be deleted from this sentence.

3. In the caption for Figure S5 "cleaved" is repeated twice in the description of (D).

Reviewer #2 (Remarks to the Author):

The authors have addressed my original concerns. However, two points in the new experiments should be clarified. 1) It is not clear what FAM-tet and FAM-Flex are; these sequences should be given if the authors claim that base composition is important - without the sequences the statement on page 6 cannot be evaluated.

2) The use of partitioning coefficient to describe FAM-tet and FAM-Flex vs. FAM-substrate is confusing, because the argument regarding length of sequences (Fig. S12) uses a diffusion measurement, making it difficult to integrate these findings. In other words, it seems possible that the length dependence is a secondary effect of the sequence dependence as both dependencies appear to be substantial. The relationship between the information gathered in these two techniques should be explained.

Reviewer #3 (Remarks to the Author):

I had two major concerns: the novelty of this study and the selective retention of a long RNA in coacervate. The second concern was appropriately addressed in the revised manuscript.

About the first concern, the authors wrote in the revised manuscript, "In contrast to these enzymes, structurally simple ribozymes, which are thought to have played a key role during early biology, lack structural stability and therefore may be rendered inactive by interactions within the highly charged and crowded interior."

I understood the novelty of this study. But I still cannot understand the reason why the authors insist that the ribozyme lacks the structural stability of a protein enzyme. Literature to support their claim should be cited. If this point is appropriately addressed, I agree to publish this article.

Thank you for your letter concerning the revision of our manuscript "Compartmentalized RNA catalysis in membrane – free coacervate protocells" by Drobot *et al.* (NCOMMS-18-06329B). We have now revised the manuscript in accord with the referees' comments and added a detailed response to the reviewers' comments and suggestions below.

We have made additional amendments to the materials and methods in line with the editorial checklist to include batch numbers for the filter sets used for wide field optical microscopy. These amendments are shown in green highlights on....On page 15, lines 391-394.

Reviewer #1 (Remarks to the Author):

I appreciate the obvious effort and attention to detail that went into the revised manuscript. My only remaining question relates to the observed differences between ribozyme concentration, and thus activity in bulk coacervates vs. a microdroplet dispersion. My sense is that the authors have tried to lay out a possible explanation as to why these differences have been observed. However, a lack of clarity in methodology and the way in which these explanations are presented seem to suggest an interesting observation, rather than a potential flaw in the experiment.

My confusion begins on page 4, where the authors state, "The final concentration of enzyme and substrate in the microdroplet dispersion was equivalent to the final concentration of the bulk coacervate phase under single turnover conditions (1 μM of HH-min and 0.5 μM FRET-substrate)." However, on the following page, they claim that the concentrations in the two phases are in fact different: "From the partition coefficient we calculated concentrations of 49.6 μM HH-min and 24.3 μM substrate in a single microdroplet compared to 1 μM HH-min and 0.5 μM substrate within the bulk coacervate phase." How is this possible? Coacervates are known to be equilibrium phases. Therefore, whether the material is present as a droplet or as a larger bulk phase, it should not be expected to have a different concentration.

Related to this, I was unclear as to how the samples were prepared. In the manuscript, the text suggests that the droplet dispersions were created by resuspending a bulk coacervate. However, there is no discussion of how this resuspension was achieved in the methods section for the activity experiments. Instead, there is a description of how an RNA-containing solution was mixed with either bulk coacervate or a dispersion of droplets. While care was taken to assure that the total volume of coacervate was the same between these two samples, there is no discussion about potential differences in equilibration time that would be critical based on the different geometries (and thus diffusion distances) between these two samples. I am very concerned that the observation of higher concentration and thus activity in the microdroplets is due to differences in the ability of

the two samples to equilibrate because of differences in geometry, rather than an intrinsic difference between the two states of the material.

In contrast, the partitioning experiments allow equilibration of RNA with a suspension of microdroplets for 10 minutes, followed by centrifugation to form a bulk coacervate phase that is assayed. Based on this procedure, I do not see how there could be a difference in the partitioning between droplets and bulk phase as both materials are used in the assay.

The only possible way in which I could envision a difference in RNA concentration between bulk and droplet materials would be if the RNA selectively binds to the droplet surface. I was unsure whether the fluorescence images shown in Figures 3 and 4 were confocal or wide-field. However, confocal imaging would clearly highlight if there were such a concentration of RNA at the surface. Additionally, I would be surprised if the concentration of the droplet could be increased by 50x in such a manner.

The authors must clarify this point in the text. Additionally, it would be possible to test for differences in equilibration by assaying the concentration of RNA in the supernatant above a bulk coacervate material as a function of time.

We thank the referee for their comments and agree that the reasoning behind the increase in RNA concentration within the coacervate microdroplets compared to the bulk coacervate phase is unclear in the materials and methods. This has led to the confusion in the main text.

The referee is correct that the coacervates are equilibrium phases and for the same volumes of coacervate there would be no difference in concentration. The reason for the discrepancy in the concentration of RNA in the droplets compared to the bulk coacervate phase is attributed to the difference in total coacervate phase volumes in the microdroplet dispersion and the bulk coacervate samples. The bulk phase sample consists entirely of the condensed coacervate phase, so 50 μL of coacervate contains 50 pmol of HH-min and 25 pmol of substrate. To produce the microdroplet sample, 1 μL of coacervate phase containing 50 pmol of HH-min and 25 pmol of substrate is dispersed in 49 μL buffer. The overall concentration of RNA is calculated from the total sample volume (*i.e.* 50 μL bulk coacervate phase and 50 μL microdroplet dispersion), so is the same in both sample types. Despite containing the same amount of RNA, the microdroplet dispersion sample contains 50x less coacervate phase than the bulk phase sample. This means that in the microdroplet dispersion, all of the RNA is strongly partitioned into only 1 μL of coacervate phase (now droplets) leading to the observed 50x increase in concentration.

The referee also raises a valid point regarding how equilibration times might affect the concentration. For the partitioning experiments, the RNA loaded bulk coacervate phase was produced by mixing of a concentrated RNA solution with the bulk coacervate phase, equilibration to allow partitioning, then centrifugation to separate the supernatant from the coacervate phase. Microdroplet samples were produced by taking the already RNA-loaded coacervate bulk phase and dispersing in supernatant. In both cases RNA is loaded only into the bulk phase, so differences in activity are not due to differences in the ability of the two samples geometries to equilibrate. In the kinetic measurements, ribozyme loaded coacervate samples are produced as described above, but the reaction is initiated by the addition of substrate to the sample, which must equilibrate into the coacervate phase. To ensure RNA substrate was distributed in the bulk coacervate phase, the samples were mechanically mixed and the dead time between the sample preparation and the first measurement was accounted for. FRAP data (Figure S7) shows that the substrate recovers in less than 20 seconds in the bulk phase, which is less than the dead time of the experiment. For the microdroplet sample, the substrate is added to the prepared microdroplet dispersion with gentle mixing. Our FRAP experiments tell us that the diffusion of the substrate into the microdroplets is fast (less than 100 secs, Figure S8a). This is again less than the dead time of the experiment.

Regarding the possibility of selective RNA binding to the droplet surface, confocal microscopy images of coacervate microdroplets loaded with tagged RNA show a homogeneous distribution of RNA through the droplet (see SI figures S8 and S12), and therefore rule out surface effects. This supports our argument that differences in RNA concentration are explained by differing volumes of coacervate phase in the two sample types in combination with the RNA partitioning coefficient.

To clarify the reasoning behind the concentration difference and preparation of sample, additional text has been added to the main text and the materials and methods:

Materials and methods: Page 11-12, line 295-308

“made up to 5 ml with nuclease free water. To produce a polymer only bulk coacervate phase (approximately 100 μ l), the aqueous dispersion of microdroplets was centrifuged (10 min, 4000 g) and the supernatant removed. To produce bulk coacervate phase containing RNA, RNA was added directly to 80 μ l of bulk coacervate phase. The samples were mechanically mixed and centrifuged (10 min, 4000 g) and any excess water from the RNA stock solutions were removed. To produce coacervate microdroplet dispersions containing the same total concentration of RNA compared to bulk coacervate phase, 1 μ l of bulk coacervate phase loaded with RNA (e.g. 50 pmol HH-min) was prepared as previously described. This 1 μ l of the bulk coacervate loaded with RNA was made up to 50 μ l with supernatant and vortexed to produce a dispersion of RNA loaded coacervate microdroplets. Final concentrations of RNA are calculated from the total volume i.e. bulk coacervate samples contain 50 μ l of bulk coacervate phase whilst the volume of the microdroplet dispersions accounts for both the volume of coacervate bulk phase and the supernatant it is dispersed in. Therefore, whilst the total concentrations are equivalent there is approximately 50x less coacervate phase within the dispersion (1 μ l) compared to the bulk coacervate phase (50 μ l).”

Main text: Page 14, 384-387

“prepared from 1 μ l of bulk coacervate phase loaded with 50 pmol HH-min in 49 μ l of supernatant. Final concentration under single turnover conditions were 1 μ M for HH-min and 0.5 μ M FRET-substrate. Dispersions were”....

Main text: Page 5, line 111-113

“formed from 1 μ l of bulk coacervate phase loaded redispersed in 49 μ l of supernatant was equivalent to the final concentration of the bulk coacervate phase i.e. 50 μ l of phase”

And: Page 5, line 124-126

“1 μ l of bulk coacervate phase was used to prepare coacervate microdroplet suspensions in a total volume of 50 μ l compared to 50 μ l of bulk coacervate phase for bulk experiments. Thus, based on the measured partition coefficients, we calculated concentrations of 49.6 μ M HH-min and 24.3 μ M substrate in a single microdroplet, compared to 1 μ M HH-min and 0.5 μ M substrate within the bulk coacervate phase.”

Minor Comments:

1. Please specify in the caption whether the fluorescence micrographs in Figures 3 and 4 from confocal or wide-field imaging.

This has been specified in the figure captions Page 29, Line 697 and page 30, line 712

2. In the methods section on page 17 related to fluorescence partitioning experiments, the text reads “to achieve a final initial concentration of 0.5 μM .” I believe that “final” should be deleted from this sentence.

We thank the reviewer for this and have removed final.

3. In the caption for Figure S5 “cleaved” is repeated twice in the description of (D).

The additional word has been removed.

Reviewer #2 (Remarks to the Author):

The authors have addressed my original concerns. However, two points in the new experiments should be clarified.

- 1) It is not clear what FAM-tet and FAM-Flex are; these sequences should be given if the authors claim that base composition is important - without the sequences the statement on page 6 cannot be evaluated.

Yes, the reviewer is correct, the sequences were already included in the SI but require additional referencing in the main text, which we have now added. We have added additional references to the table which contain the sequences and an additional explanation of the sequences in the main text Page 6 and line 160-167

“To investigate additional sequence-dependent effects on partitioning, we compared the partition coefficients of different 12mer RNAs: FAM-substrate is pyrimidine rich but unstructured RNA; FAM-tet is a pyrimidine-rich hairpin structure with a stable UUCG tetraloop; FAM-flex is an unstructured purine-rich sequence (see Table S2). Our results show that for unstructured RNAs an increase in purine content reduces partitioning of 12mer RNAs (FAM-substrate vs FAM-flex) approximately 10-fold. Likewise, we observe a decrease in the partition coefficient with an increase in secondary structure for pyrimidine-rich RNA (FAM-substrate vs FAM-tet) (Figure S9).”

And to the figure caption in Figure S9, SI page 11, lines 187-189

“Figure S9: Partitioning (K) of three different sequences RNA (i) the unstructured pyrimidine-rich 12-mer FAM-substrate (ii), the unfolded, purine-rich 12-mer FAM-Flex (iii) and the stable, pyrimidine hairpin structure 12-mer FAM-Tet. Sequences are shown in Table S2”

- 2) The use of partitioning coefficient to describe FAM-tet and FAM-Flex vs. FAM-substrate is confusing, because the argument regarding length of sequences (Fig. S12) uses a diffusion measurement, making it difficult to integrate these findings. In other words, it seems possible that the length dependence is a secondary effect of the sequence dependence as both dependencies appear to be substantial. The relationship between the information gathered in these two techniques should be explained.

We thank the reviewer for this comment and agree that the dependence of the partition coefficient on both the length and sequence is substantial. We also agree that these two effects are likely coupled as e.g. also shown by work of Aumiller Jr *et al.* (Langmuir, 2016, 32, 10042–10053), which we cite in our manuscript. We also agree that there could be some confusion with regard to the use of partition coefficients in one part to describe the length dependence of the partitioning coefficient, and then the FRAP experiments (diffusion based) to describe the affinity of the RNA to the PLys-ATP. In fact, the FRAP experiments are complementary to the partition experiments and tell us about the rate of exchange between the aqueous environment and the coacervate environment, whilst the partition coefficients give a measure of the equilibrium state of RNA distribution between the coacervate phase and the aqueous environment. A similar strong correlation with the FRAP recovery half-life and the partitioning coefficient was previously shown by Aumiller Jr *et al.* To make this correlation clearer to the reader, additional text was added to describe the correlation between the partition coefficients and the FRAP data (page 6, line 151-159):

“attributed to either a low concentration of HH-min within the surrounding aqueous phase; a slow rate of transfer into the coacervate droplet from its exterior and/or immobilized ribozyme in the coacervate droplet which is unable to exchange with RNA in the surrounding aqueous phase (Figure S8). The results from the FRAP-experiments complement the equilibrium partition coefficient: While the 12-mer substrate with a lower partition

coefficient ($K=3000\pm 2000$) shows a faster exchange between the droplet and the surrounding aqueous phase compared, the 39-mer ribozyme, which has a higher partition coefficient ($K = 9600 \pm 5600$) shows slower exchange of RNA with the surrounding environment. A strong correlation between FRAP half-lives and partitioning coefficients was also described for RNAs in other coacervate systems⁴.”

In addition, we have included the partition coefficient data of the different length RNA for the PLys:ATP system as an additional figure into the SI and referenced this data in the main text by the following additions on page 8, line 207-209:

“Both the partition coefficient and whole droplet FRAP results for PLys : ATP coacervate microdroplets, containing either TAM-HH-min (39-mer), FAM-substrate (12-mer) or cleaved FAM-substrate (6-mer) (Figure S12 and Figure S13),”

Additional figure as new Figure S13 in the SI Page 15.

Figure S13: Partitioning of three different length RNA (i) 39-mer TAM-HH-min, (ii) 12-mer FAM-substrate, (iii) 6-mer FAM-Cleaved-substrate and (iv) FITC (v) TAMRA in the 1:4 (final molar ratio) ATP : PLys coacervate system (A-C). To obtain the partition coefficient $0.36 \mu\text{M}$ (total concentration) of either dye tagged RNA or dye, each sample was incubated in $150 \mu\text{L}$ ATP / PLys coacervate microdroplet solution ($3 \mu\text{L}$ bulk coacervate phase + $147 \mu\text{L}$ supernatant) and left to equilibrate for 10 min. The dispersion was then centrifuged and the supernatant separated from the coacervate phase. Either $3 \mu\text{L}$ of the bulk coacervate phase or $147 \mu\text{L}$ supernatant was made up to $150 \mu\text{L}$ by the addition of $147 \mu\text{L}$ RNA free supernatant or $3 \mu\text{L}$ RNA free coacervate, respectively. The fractions of dye tagged RNA or dye in the coacervate phase (A) and in the supernatant phase (B) were obtained spectroscopically. Partitioning coefficients were calculated using Equation 9-11 and plot for the ATP : PLys coacervate system (C) with the y-axis plot on the logarithmic scale. The results show the general trend that longer length RNA partitions more into the coacervate phase compared to shorted length RNA with (i) 39-mer TAM-HH-min ($K = 8000 \pm 4000$), (ii) 12-mer FAM-substrate ($K = 2700 \pm 700$), (iii) 6-mer FAM-Cleaved-substrate ($K = 1400 \pm 300$). The partition coefficients for (iv) FITC ($K = 177 \pm 12$), (v) TAMRA ($K = 120 \pm 6$) are an order of magnitude less than the dye tagged RNA showing that partitioning is primarily driven by the nucleotide and not the dye molecule.

Finally, we have now re-written the paragraph on page 6, lines 160-169, to further emphasize that both RNA length and sequence have a considerable effect on partitioning:

“To investigate additional sequence-dependent effects on partitioning, we compared the partition coefficients of different 12mer RNAs (Figure S6): FAM-substrate is pyrimidine rich but unstructured RNA; FAM-tet is a pyrimidine-rich hairpin structure with a stable UUCG tetraloop; FAM-flex is an unstructured purine-rich sequence (see Table S2). Our results show that for unstructured RNAs an increase in purine content reduces partitioning of 12mer RNAs (FAM-substrate vs FAM-flex) approximately 10-fold. Likewise, we observe a decrease in the partition coefficient with an increase in secondary structure for pyrimidine-rich RNA (FAM-substrate vs FAM-tet) (Figure S9). Thus, our results, and others,^{4,42,43} show that RNA sequestration and localization within coacervate droplets is dependent on the length, sequence and also structure of the sequestered RNA.”

Reviewer #3 (Remarks to the Author):

I had two major concerns: the novelty of this study and the selective retention of a long RNA in coacervate. The second concern was appropriately addressed in the revised manuscript.

About the first concern, the authors wrote in the revised manuscript, "In contrast to these enzymes, structurally simple ribozymes, which are thought to have played a key role during early biology, lack structural stability and therefore may be rendered inactive by interactions within the highly charged and crowded interior."

I understood the novelty of this study. But I still cannot understand the reason why the authors insist that the ribozyme lacks the structural stability of a protein enzyme. Literature to support their claim should be cited. If this point is appropriately addressed, I agree to publish this article.

We thank the reviewer for this comment. We agree that the term "difference in structural stability" in terms of comparing RNA vs protein requires a little more explanation. Firstly, protein enzymes typically have globally compact structures, which protect and stabilize the active site. In comparison, simple (small) ribozymes typically have an open and flexible structure, which is more susceptible to alterations in folding by external factors such as the presence of electrostatic charges (e.g. Mg^{2+}). Secondly, RNA-folding is characterised by a much more rugged energy landscape compared to the protein folding landscape (Chen & Dill, (2000). *Proc. Natl. Acad. Sci. U. S. A.* **97**, 646–65). This often leads to populations of heterogeneous RNA folding intermediates separated by high energetic barriers from the native structure (Russel, (2008) *Front Biosci.*; **13**: 1–20). *In vivo*, RNA-chaperones assist folding of biologically active RNAs into their native structure, which are absent *in vitro* or in a hypothetical early RNA world. Finally, RNA folding is strongly dependent on solving the "polyelectrolyte problem" (repulsion based on the negatively charged phosphate backbone) before folding can proceed via Watson-Crick base pairing and base stacking (Thirumalai and Hyeon, 2005, *Biochemistry*, **44**, 4957-4970). In contrast, protein folding is initiated by a "collapse" leading to the burial of hydrophobic residues. Thus, it is conceivable that RNA folding and, thus ribozyme activity, is much more influenced by the highly charged environment inside coacervates. Indeed, the detrimental impact of the coacervate environment on hammerhead folding is directly visible in Figure S4 of our manuscript.

To clarify this, we have added a citation to the text to support this claim and modified the sentence on page 2, line 50-53 as follows:

"In contrast to these enzymes, structurally simple ribozymes, which are thought to have played a key role during early biology, are prone to fold into inactive conformations in the absence of RNA chaperones or additional auxiliary elements^{29–33} and therefore may be rendered inactive by interactions within the highly charged and crowded interior of coacervate microdroplets."

With additional references:

29. Chen, S.-J. & Dill, K. A. RNA folding energy landscapes. *Proc. Natl. Acad. Sci. U. S. A.* **97**, 646–651 (2000).
30. Thirumalai, D. & Hyeon, C. RNA and Protein Folding: Common Themes and Variations. *Biochemistry* **44**, 4957–4970 (2005).
31. Russell, R. RNA misfolding and the action of chaperones. *Front. Biosci. J. Virtual Libr.* **13**, 1–20 (2008).

32. Penedo, J. C., Wilson, T. J., Jayasena, S. D., Khvorova, A. & Lilley, D. M. J. Folding of the natural hammerhead ribozyme is enhanced by interaction of auxiliary elements. *RNA* **10**, 880–888 (2004).
33. Gago, S., De la Peña, M. & Flores, R. A kissing-loop interaction in a hammerhead viroid RNA critical for its in vitro folding and in vivo viability. *RNA* **11**, 1073–1083 (2005).

REVIEWERS' COMMENTS:

Reviewer #1 (Remarks to the Author):

I would like to thank the authors for the clarifications that they have incorporated into the new version of the manuscript. These assist in understanding the experimental setup and results. The only other comment I might make in regards to the observed differences in the kinetic rate constant between the bulk coacervate and the microdroplets is that the different concentrations of RNA present in the two materials might result in different levels of carboxymethyl dextran, which might also account for the observed differences. This possibility is just one alongside the others already raised by the authors. Aside from several other minor comments, I am satisfied with the manuscript in its current form and recommend it for publication.

Minor Comments:

1. I would suggest specifying that the phenomenon described here is "complex coacervation." This would only need to be done the first time that the concept is mentioned, and possibly in the abstract. "Coacervation" would be sufficient from there on out.
2. At the bottom of page 4, the text reads "1 μ l of bulk coacervate phase loaded redispersed..." Is perhaps a word missing here?

Reviewer #2 (Remarks to the Author):

My concerns were addressed.

Thank you for your letter concerning the revision of our manuscript “Compartmentalised RNA catalysis in membrane-free coacervate protocells” by Drobot et al. (NCOMMS-18-06329C). We are pleased to see that we could address all major concerns of the referees during the second round of revision. Please find below our response to the remaining minor comments from Reviewer #1.

RESPONSE TO REVIEWER:

Reviewer #1 (Remarks to the Author):

I would like to thank the authors for the clarifications that they have incorporated into the new version of the manuscript. These assist in understanding the experimental setup and results. The only other comment I might make in regards to the observed differences in the kinetic rate constant between the bulk coacervate and the microdroplets is that the different concentrations of RNA present in the two materials might result in different levels of carboxymethyldextran, which might also account for the observed differences. This possibility is just one alongside the others already raised by the authors. Aside from several other minor comments, I am satisfied with the manuscript in its current form and recommend it for publication.

Minor Comments:

1. I would suggest specifying that the phenomenon described here is “complex coacervation.” This would only need to be done the first time that the concept is mentioned, and possibly in the abstract. “Coacervation” would be sufficient from there on out.

We thank the referee for this suggestion. We have now introduced the term “complex coacervation” in the abstract and the introduction.

“ABSTRACT

*Phase separation of mixtures of oppositely charged polymers provides a simple and direct route to compartmentalization via **complex** coacervation, which may have been important for driving primitive reactions as part of the RNA world hypothesis. However, to date, RNA catalysis has not been reconciled with coacervation. Here we demonstrate that RNA catalysis is viable within coacervate microdroplets and further show that these membrane-free droplets can selectively retain longer length RNAs while permitting transfer of lower molecular weight oligonucleotides.”*

“Introduction

*Compartmentalization driven by spontaneous self-assembly processes is crucial for spatial localisation and concentration of reactants in modern biology and may have been important during the origin of life. One route known as **complex** coacervation describes a complexation process^{1,2} between two oppositely charged polymers such as polypeptides and nucleotides.³⁻⁷ The resulting coacervate microdroplets are membrane free, chemically enriched and in dynamic equilibrium with a polymer poor phase. In addition to being stable over a broad range of physicochemical conditions, coacervate droplets are able to spatially localize and up-concentrate different molecules^{3,8} and support biochemical reactions.^{9,10”}*

2. At the bottom of page 4, the text reads “1 µl of bulk coacervate phase loaded redispersed...” Is perhaps a word missing here?

Indeed, the word "loaded" was superfluous in this sentence and we have now removed it. The sentence now reads:

“The final concentration of enzyme and substrate in the microdroplet dispersion, formed from 1 µl of bulk coacervate phase redispersed in 49 µl of (...)”

All changes have been implemented into the marked up manuscript with track changes.